# Teasing out missing reactions in genome-scale metabolic networks through hypergraph learning

Can Chen [1,4], Chen Liao [2,4] & Yang-Yu Liu [1,3] ✉

GEnome-scale Metabolic models (GEMs) are powerful tools to predict cellular metabolism and physiological states in living organisms. However, due to our imperfect knowledge of metabolic processes, even highly curated GEMs have knowledge gaps (e.g., missing reactions). Existing gap-filling methods typically require phenotypic data as input to tease out missing reactions. We still lack a computational method for rapid and accurate gap-filling of metabolic networks before experimental data is available. Here we present a deep learning-based method – CHEbyshev Spectral HyperlInk pREdictor (CHESHIRE) – to predict missing reactions in GEMs purely from metabolic network topology. We demonstrate that CHESHIRE outperforms other topology-based methods in predicting artificially removed reactions over 926 high- and intermediate-quality GEMs. Furthermore, CHESHIRE is able to improve the phenotypic predictions of 49 draft GEMs for fermentation products and amino acids secretions. Both types of validation suggest that CHESHIRE is a powerful tool for GEM curation to reveal unknown links between reactions and observed metabolic phenotypes.

As a mathematical representation of the metabolism for an organism, the GEnome-scale Metabolic model (GEM) offers a comprehensive gene-reaction-metabolite connectivity through two matrices: the stoichiometric matrix associating metabolites with their reactions; and the reaction-gene matrix associating reactions with their corresponding enzymes and genes[1,2]. GEMs are powerful computational tools to predict metabolic fluxes in living organisms[2,3]. Used alone or integrated with high-throughput data, GEMs can produce mechanistic insights and falsifiable predictions that progressively advance various disciplines in biomedical sciences[4,5], including metabolic engineering[6,7], microbial ecology[8], and drug discovery[9]. Recently, the rapid growth in whole-genome sequencing data[10] has triggered a surge in draft GEMs generated by automatic reconstruction pipelines[11,12]. Yet, these draft models contain knowledge gaps and thus require comprehensive manual curation[13,14], e.g., finding missing reactions due to

incomplete genomic and functional annotations. Therefore, the quality of initial draft GEMs has a profound impact on the time spent on the manual curation, the refined model quality, and ultimately its utility in biomedical applications.

Numerous optimization-based gap-filling methods have been designed to tease out missing reactions in draft GEMs[15–18]. Despite wide differences in their input data types, objectives, and algorithms, they generally follow two steps: (1) find dead-end metabolites that cannot be produced or consumed and/or some inconsistencies between the draft model prediction and experimental data (e.g., growth profiles); and (2) add a set of reactions to resolve the dead-end blocks and/or inconsistencies[16]. Optimization-based methods often require data as input to identify model simulation-data inconsistencies[15,16]. However, experimental data is not readily available for non-model organisms, thus limiting the utility of those tools. For example, most intestinal

[1]Channing Division of Network Medicine, Department of Medicine, Brigham and Women's Hospital and Harvard Medical School, Boston, MA 02115, USA. [2]Program for Computational and Systems Biology, Memorial Sloan Kettering Cancer Center, New York, NY 10065, USA. [3]Center for Artificial Intelligence and Modeling, The Carl R. Woese Institute for Genomic Biology, University of Illinois at Urbana-Champaign, Champaign, IL 61801, USA. [4]These authors contributed equally: Can Chen, Chen Liao. ✉e-mail: yyl@channing.harvard.edu

organisms are considered "uncultivable" and their functions remain unknown[19]. Even for cultivable organisms, high-throughput phenotypic screening, i.e., searching for organisms with desired phenotypes, relies on the analysis of microbial extracts or genetic modifications, which can become complicated, time-consuming, and expensive. Given the increasing availability of cultivable organisms and their genomes, there is a pressing need for rapid and accurate in silico predictions of metabolic phenotypes solely from genomic sequences. Even though the predictions are theoretical, downstream experimental validations could be mush less resource-demanding.

A few gap-filling methods that are entirely topology-based and do not require phenotypic data as input include (1) classical methods that restore the network connectivity based on flux consistency, such as GapFind/GapFill[20] and FastGapFill[21]; and (2) state-of-the-art machine learning methods that exploit advanced machine learning techniques, such as Neural Hyperlink Predictor (NHP)[22] and Clique Closure-based Coordinated Matrix Minimization (C3MM)[23] (see "Supplementary Note 1"). Machine learning methods frame the prediction of missing reactions in a GEM as a task of predicting hyperlinks on a hypergraph[22–24]. Compared to graphs where each link connects two nodes, hypergraphs allow each hyperlink to connect more than two nodes[25–28] (see "Hypergraphs in Methods"). Notably, metabolic networks or any biochemical reaction networks have a very natural hypergraph representation: each molecular species is a node and each reaction is a hyperlink connecting all the molecular species involved in it.

There are notable limitations associated with existing topology-based machine learning methods. C3MM has an integrated training-prediction process, which includes all candidate reactions (obtained from a reaction pool) during training. Hence, it has limited scalability (i.e., it cannot handle large reaction pools), and the model has to be re-trained for each new reaction pool. While the neural network-based method NHP separates candidate reactions from training, it approximates hypergraphs using graphs in generating node features, which results in the loss of higher-order information. More importantly, both methods were benchmarked against a handful of GEMs (lacking a comprehensive test) and were only internally validated using artificial gaps introduced by randomly deleting reactions from input GEMs (lacking validations on predicting metabolic phenotypes, i.e., external validation).

Here we develop a method called CHESHIRE (CHEbyshev Spectral HyperlInk pREdictor) to overcome the limitations of existing machine learning methods. CHESHIRE only requires a metabolic network for training and outputs confidence scores for candidate reactions from a reaction pool. For internal validation, we show that CHESHIRE outperforms NHP and C3MM in systematic tests of recovering artificially removed reactions from 108 BiGG[29] models and 818 AGORA[8] models. For external validation, we assess the ability of CHESHIRE to predict metabolic phenotypes. Using 49 draft GEMs reconstructed from commonly used pipelines (CarveMe[11] and ModelSEED[30]), we show that CHESHIRE improves the theoretical predictions of whether fermentation metabolites and amino acids are produced by these GEMs.

## Results

### A brief overview of CHESHIRE

CHESHIRE is a deep learning-based method that predicts missing reactions in GEMs using topological features of their metabolic networks without any inputs from experimental data (other than genomic sequences). For each metabolic network (Fig. 1a), we use a hypergraph (Fig. 1b) to model its structure, where each hyperlink represents a metabolic reaction and connects participating reactant and product metabolites. CHESHIRE takes the incidence matrix of the hypergraph and a decomposed graph (built from the hypergraph of existing or candidate reactions) as input. The former contains boolean values indicating the presence or absence of each metabolite in each reaction.

The latter consists of fully connected subgraphs (each subgraph represents a reaction with all its metabolites connected) formed by positive and negative reactions during training and by candidate reactions during prediction (Fig. 1c, d). Positive reactions are those existing in the metabolic network, while negative reactions are fake (do not exist) and created for model-balancing purposes (often referred to as negative sampling). Note that only positive reactions are used to construct the incidence matrix.

The learning architecture of CHESHIRE has four major steps: feature initialization, feature refinement, pooling, and scoring (Fig. 1e, f). For feature initialization, we employ an encoder-based one-layer neural network[31] to generate a feature vector for each metabolite from the incidence matrix (see "Feature initialization in Methods"). This initial feature vector encodes the crude information of topological relationship of a metabolite with all reactions in the metabolic network. For feature refinement, to capture the metabolite-metabolite interactions, we use Chebyshev spectral graph convolutional network (CSGCN)[32] on the decomposed graph to refine the feature vector of each metabolite by incorporating the features of other metabolites from the same reaction (see "Feature refinement in Methods"). For pooling (i.e., integrating node- or metabolite-level features into hyperlink- or reaction-level representation), we utilize graph coarsening methods to compute a feature vector for each reaction (represented by a fully connected subgraph in the decomposed graph) from the feature vectors of its metabolites. We combine two pooling functions, a maximum minimum-based function (as used in NHP[22]) and a Frobenius norm-based function[33] to provide complementary information of metabolite features. Finally, for scoring, we feed the feature vector of each reaction into a one-layer neural network to produce a probabilistic score for the reaction that indicates the confidence of its existence (see "Pooling and scoring in Methods"). In the training phase, the resulting scores are compared to the target scores (one for positive reactions and zero for negative reactions) with a loss function for updating the model parameters (Fig. 1e, see "Training algorithm in Methods"). Compared with NHP which shares a similar architecture, CHESHIRE exploits a simple encoder[31], a sophisticated CSGCN[32], and a practical Frobenius norm-based pooling function[33] (see "Difference between CHESHIRE and NHP in Supplementary Note 2"). For hyperparameters of CHESHIRE, see "Hyperparameter selection in Methods".

### Internal validation of CHESHIRE using artificially introduced gaps

The goal of internal validation is to test the ability of CHESHIRE to recover artificially introduced gaps (i.e., removing existing reactions in metabolic networks). We compared CHESHIRE with the state-of-the-art machine learning methods NHP and C3MM as they have been demonstrated to display superior performances over previous gap-filling (or hyperlink prediction) methods. We also included Node2Vec-mean[22,34] (referred to as NVM below) as a baseline method. The learning architecture of NVM is relatively simple. It uses Node2Vec (a random walk-based graph embedding method that generates node features) and mean pooling (averaging node features) to generate metabolite and reactions features, respectively, without feature refinement (see "Node2Vec-mean in Supplementary Note 1").

Below we performed two types of internal validation based on artificially introduced gaps (Fig. 2a). For both types, metabolic reactions in a given GEM were first split into a training set and a testing set over 10 Monte Carlo runs. All the deep learning-based methods (CHESHIRE, NHP, and NVM) require negative sampling of reactions. We created negative reactions at 1:1 ratio to positive reactions for the training and testing sets, respectively, by replacing half (rounded if needed) of the metabolites in each positive reaction with randomly selected metabolites from a universal metabolite pool (Fig. 2a, see "Negative sampling in Methods"). In the first type of internal validation, the training and testing sets of positive reactions and their derived

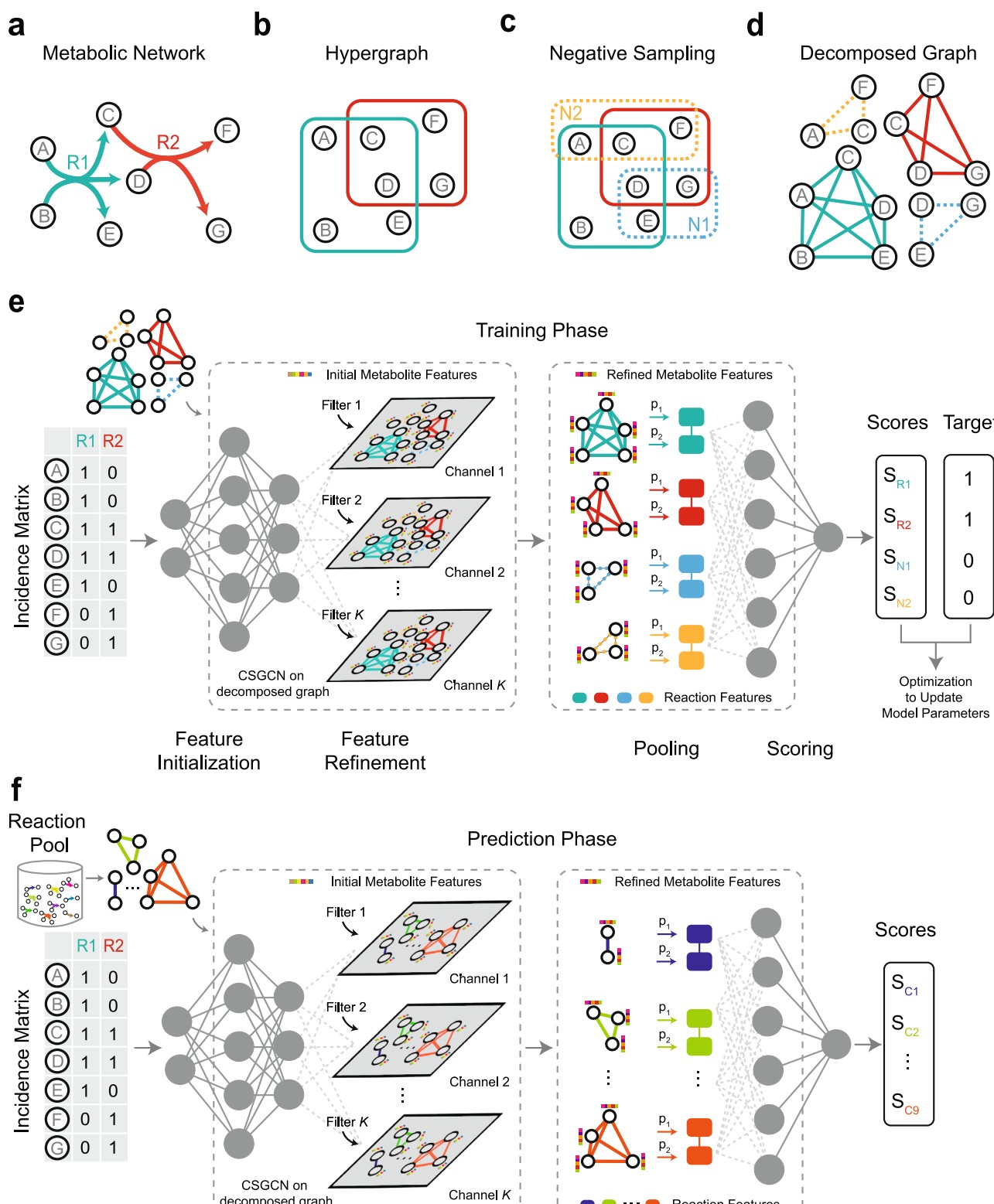

**Fig. 1 | CHESHIRE workflow. a** Schematic representation of a metabolic network.
**b** Hypergraph representation of the metabolic network. The hypergraph is
undirected where each hyperlink connects metabolites that participate the same
reaction. **c** Negative sampling of the metabolic network. Solid and dashed boxes
represent positive and negative reactions (e.g., N1, N2), respectively.
**d** Decomposed graph of the metabolic network, where each reaction (either
positive or negative) is treated as a fully connected subgraph (solid and dashed
lines represent positive and negative reactions, respectively). **e** The architecture of
CHESHIRE during training. The deep neural network takes the incidence matrix

and the decomposed graph (**d**) as input, and consists of an encoder layer, a Che-
byshev spectral graph convolutional layer with *K* filters (resulting in *K* channels), a
pooling layer with two pooling functions, and a final scoring layer. The output
confidence scores are compared to the target scores for updating model para-
meters. The gray dots represent the hidden neurons. **f** The architecture of CHE-
SHIRE during prediction. The neural network takes the incidence matrix and a
decomposed graph built from candidate reactions as input and outputs con-
fidence scores for candidate reactions based on the trained model parameters.

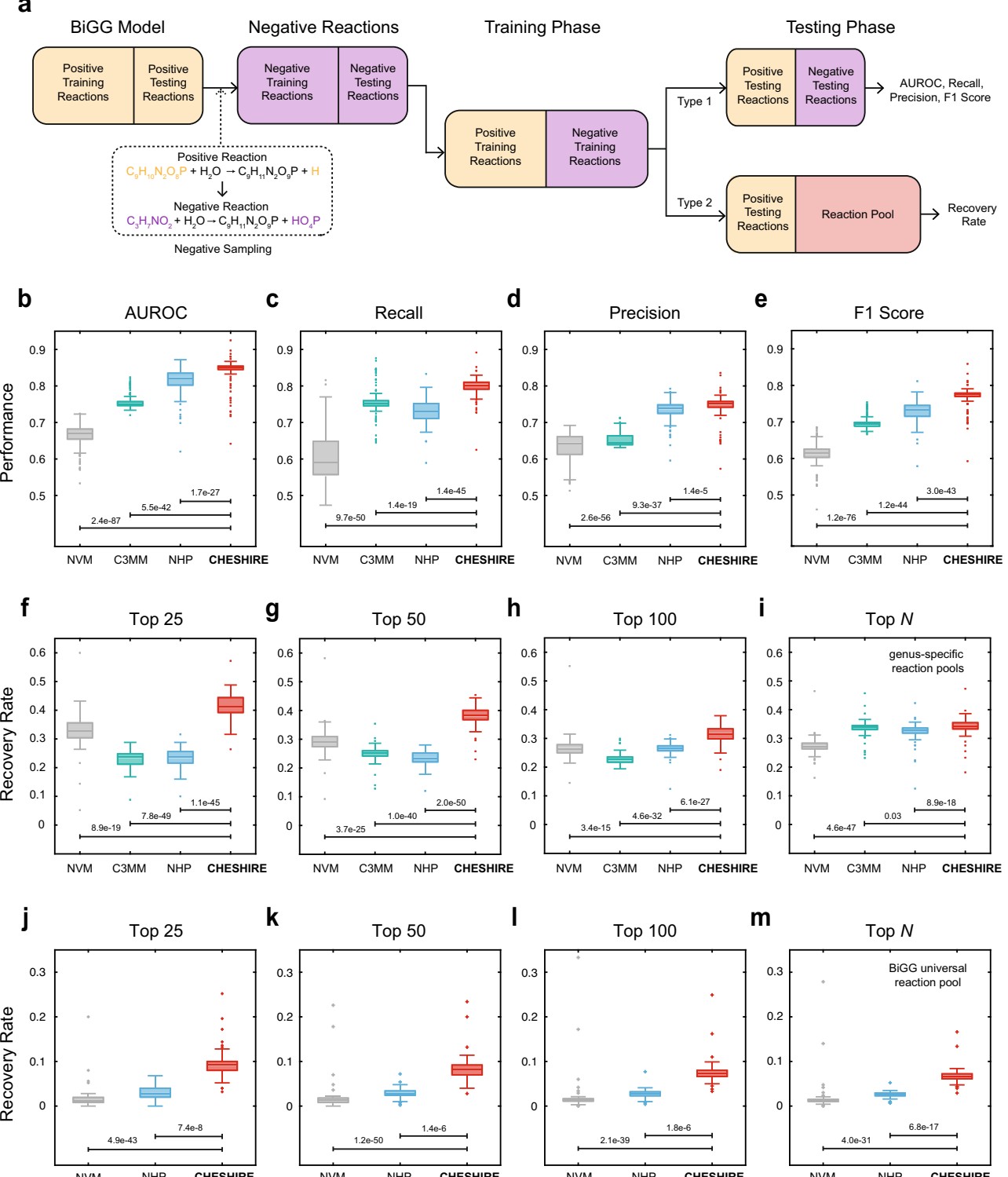

**Fig. 2 | Internal validation using artificially introduced gaps. a** Flowchart of internal validation. Two types of internal validation were performed. The former mixes artificially removed positive reactions and their derived negative reactions as candidate reactions, while the latter uses artificially removed positive reactions and real reactions from a universal reaction database as candidate reactions. **b–e** Boxplots of the performance metrics (AUROC, Recall, Precision, and F1 score) calculated on 108 BiGG GEMs (each dot represents a GEM) for CHESHIRE vs. NHP, C3MM, and NVM. **f–i** Reaction recovery rate of CHESHIRE vs. NHP, C3MM, and NVM for gap-filling the BiGG GEMs using genus-specific reaction pools. The comparison was performed on 73 BiGG models which have over 1000 reactions and whose

genera are present in the genus-specific reaction pools, by adding the top 25, 50, 100, and N reactions with the highest confidence scores (N is the number of artificially removed reactions). **j–m** The same as (**f–i**) but using the entire BiGG universal reaction pool. 83 BiGG models with over 1000 reactions were tested, and C3MM was excluded due to the issue of scalability. Each data point represents the mean statistic over 10 Monte Carlo runs. Boxplot: central line represents the median, box limits represent the first and third quartiles, and whiskers extend to the smallest and largest values or at most to 1.5× the interquartile range, whichever is smaller. Two-sided paired-sample t-test: exact p-values are provided. Source data are provided as a Source Data file.

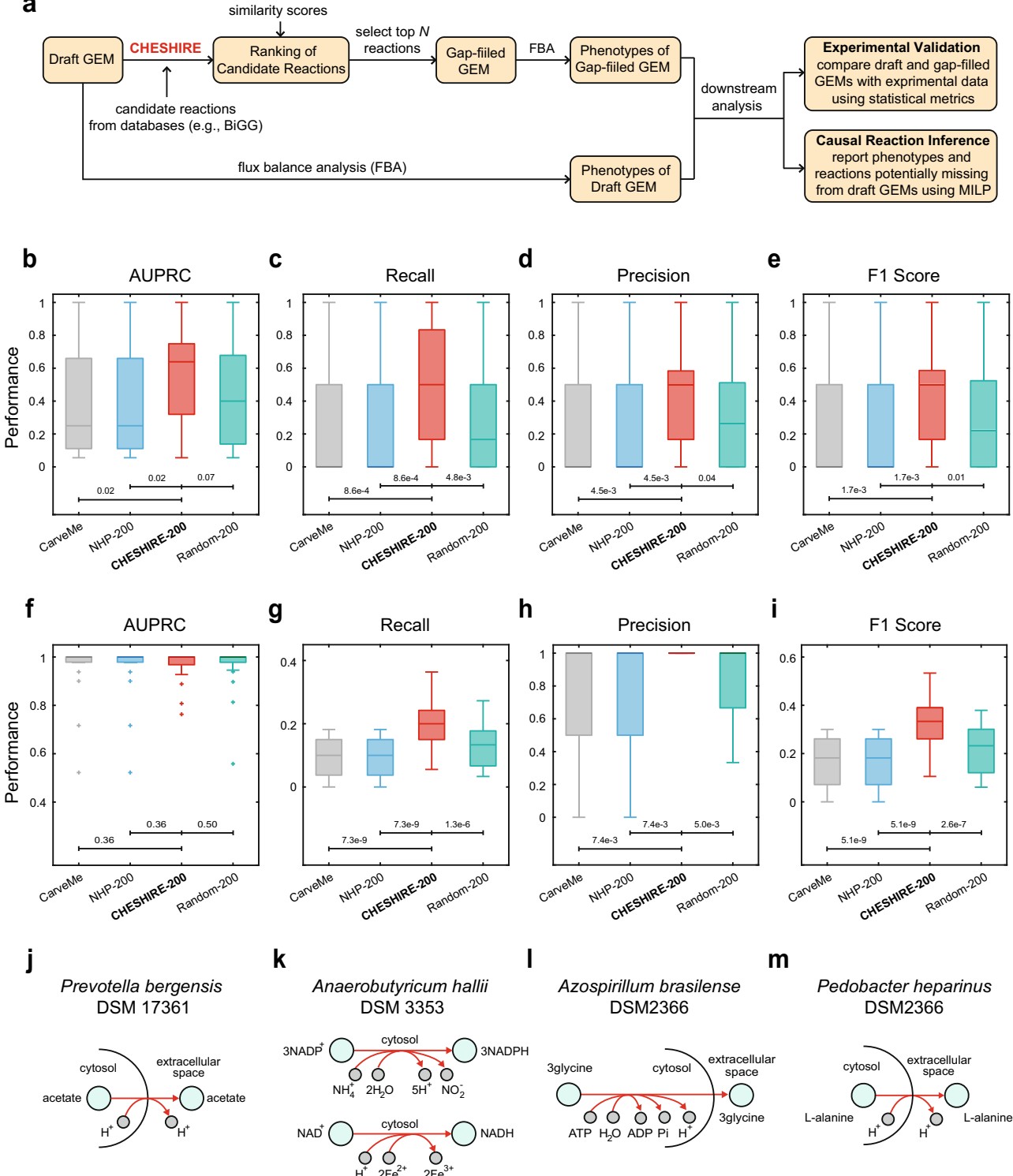

**j** *Prevotella bergensis* DSM 17361

**k** *Anaerobutyricum hallii* DSM 3353

**l** *Azospirillum brasilense* DSM2366

**m** *Pedobacter heparinus* DSM2366

negative reactions were combined and used for training and testing, respectively. For fair comparison, we also introduced negative reactions to the testing set of C3MM. In the second type of internal validation, every step remains the same except that the testing set was not mixed with its derived negative reactions but with real reactions from a universal database.

To perform the first type of internal validation, we tested CHE-SHIRE on a total of 108 high-quality BiGG GEMs (see "BiGG models in Supplementary Note 3") with 60% training and 40% testing. CHESHIRE achieves the best performance in different classification performance

metrics, including the Area Under the Receiver Operating Characteristic curve (AUROC), Recall, Precision, and F1 score (the harmonic mean of Recall and Precision) (Fig. 2b−e). A threshold score of 0.5 was used to determine whether a test reaction is true or false. Compared to the overall second-best method NHP, CHESHIRE has a significantly higher level of Precision ($P < 10^{-4}$, two-sided paired-sample $t$-test) and Recall ($P < 10^{-16}$), suggesting that CHESHIRE recovers the majority of true reactions without sacrificing its ability to distinguish true from fake reactions. For all the performance metrics, the wide distributions indicate that the performance of CHESHIRE (as well as the other three

**Fig. 3 | External validation by predicting metabolic phenotypes. a** Flowchart of external validation. The predicted phenotypes from CHESHIRE-gapfilled GEMs are validated by comparison to experimental observation. For phenotypes correctly predicted by gap-filled GEMs but missed by draft GEMs, we also identify the causal reactions from CHESHIRE-predicted set that improve the phenotypic prediction using Mixed Integer Linear Programming (MILP). **b–i** Performance (AUPRC, Recall, Precision, and F1 score) of CHESHIRE and NHP in filling gaps in (**b–e**) 24 bacterial GEMs for fermentation metabolite production and (**f–i**) 25 bacterial GEMs for amino acid secretions. NVM was not included here due to its poor performance in internal validation. C3MM was not considered either because of the issue of scalability. "CarveMe" represents the draft models reconstructed from CarveMe. "NHP-200" and "CHESHIRE-200" represent draft models plus 200 reactions predicted by NHP and CHESHIRE, respectively (reaction confidence scores averaged over 5

Monte Carlo runs). For "Random-200", 200 randomly selected reactions from the universal BiGG database were added to the draft models (performance averaged over 3 Monte Carlo runs). Boxplot: central line represents the median, box limits represent the first and third quartiles, and whiskers extend to the smallest and largest values or at most to 1.5× the interquartile range, whichever is smaller. Two-sided paired-sample $t$-test: exact $p$-values are provided. **j–m** Examples of CHESHIRE-predicted reactions (red arrows) that causally gap-fill the observed phenotypes of acetate production (**j**), lactate production (**k**), and amino acid secretions (**l**, **m**). Abbreviations of cofactors: adenosine triphosphate (ATP); adenosine diphosphate (ADP); adenosine phosphate (AMP); phosphate (Pi); inorganic pyrophosphate (PPi); Coenzyme A (CoA); oxidized/reduced nicotinamide adenine dinucleotide (NAD$^+$/NADH); oxidized/reduced nicotinamide adenine dinucleotide phosphate (NADP$^+$/NADPH). Source data are provided as a Source Data file.

methods) is GEM-dependent, even though they are all high-quality models. The most easily gap-filled GEM is a reconstruction from *Phaeodactylum tricornutum* CCAP 1055/1 (model iLB1027_lipid), where CHESHIRE is able to achieve AUROC = 0.92, Recall = 0.89, Precision = 0.82, and F1 score = 0.86 on the testing set. Notably, CHESHIRE is not sensitive to the threshold score, the negative sampling strategy, and the negative sampling ratio in this validation, where its performance still prevails over the other methods (Supplementary Fig. 1–3, see "Threshold scores, Negative sampling strategies, and Negative sampling ratios in Supplementary Notes 4"). We also performed the same type of internal validation on a larger set of metabolic networks, i.e., 818 AGORA models from the Virtual Metabolic Human database[8], and observed similar results (Supplementary Fig. 4).

To perform the second type of internal validation, we tested CHESHIRE on the same BiGG GEMs with 90% training and 10% testing using genus-specific and the universal BiGG reaction pools (see "Construction of BiGG genus-specific reaction pools and Construction of BiGG universal reaction pool in Supplementary Note 3"). The former has a relatively small size (200–800 reactions) per GEM, while the latter has almost 17,000 reactions. Since the number of reactions with similar biochemistry mechanisms and thus nearly identical confidence scores scale up with the size of candidate reaction pool, a loose threshold of 0.5 may still predict hundreds or thousands of candidates as missing reactions. Instead of using a fixed cutoff threshold, we added the top 25, 50, 100, and $N$ reactions with the highest confidence scores ($N$ is the number of artificially removed reactions). We found that CHESHIRE achieves the highest recovery rate at the four cutoffs for both types of reaction pools (Fig. 2f–m). Notably, by adding the top 25 reactions from the genus-specific reaction pools, CHESHIRE identifies more than 40% (on average) artificially removed reactions, significantly outperforming the other three methods (Fig. 2f; $P < 10^{-16}$, two-sided paired-sample $t$-test). While the performance of CHESHIRE declines when more reactions were added, it is still significantly better than the second-best method C3MM at the top $N$ cutoff (Fig. 2i; $P < 0.05$). Furthermore, as expected, using the entire BiGG universal reaction pool would undermine the performances of recovery rate for all the methods. CHESHIRE nevertheless accomplishes the best performance compared to NHP and NVM (Fig. 2j–m; $P < 10^{-5}$).

Taken together, the two types of internal validations demonstrate that CHESHIRE outperforms other topology-based methods and hence is more promising for predicting missing reactions in draft GEMs.

### External validation of CHESHIRE via phenotypic prediction

Compared to internal validation that tests the predictions by using artificially removed reactions as the ground truth, external validation tests whether gap-filled GEMs by CHESHIRE has improved performance compared to draft GEMs in terms of their predictions of phenotypic data (Fig. 3a, see "Supplementary Note 5"). This test is biologically meaningful. After all, a major rationale for reconstructing GEMs of microorganisms is to provide theoretical predictions of their metabolic phenotypes[35].

Briefly, CHESHIRE is trained on the entire reaction set of a draft GEM, and candidate reactions (taken from a reaction pool, e.g., the BiGG database[29]) are ranked based on both confidence and similarity scores. The confidence score, returned by CHESHIRE, quantifies the probability of a candidate reaction being present in the GEM. The similarity score measures the maximum correlation between a candidate reaction and all existing reactions in the GEM. Given our rationale that dissimilar reactions are more likely to be functionally complementary to the existing ones, all candidate reactions whose confidence scores ≥ 0.9995 are ranked by their similarity scores (least similar to most similar). The top 200 reactions are added to the draft GEM to produce a gap-filled GEM (see "Generation of GEMs in Supplementary Note 5"). Particularly, any reaction causing energy-generating cycles (EGCs)[36] is included if EGCs can be eliminated by changing its flux bounds and otherwise skipped. Given a culture medium (see "Culture media compositions in Supplementary Note 5"), the simulated phenotypes of both draft and gap-filled GEMs (see "Simulations of metabolic phenotypes in Supplementary Note 5") are then compared to experimental observation for validation.

We applied the workflow to a compiled dataset including fermentation profiles of 9 metabolites from 24 bacterial organisms (Supplementary Table 1, see "Fermentation metabolite test data in Supplementary Note 3") grown under anaerobic conditions[12]. The draft GEMs of those organisms were reconstructed using a recent automatic reconstruction pipeline CarveMe[11]. The same set of performance metrics as used in the internal validation was used, except that AUROC was replaced by AUPRC (the Area Under Precision-Recall Curve) for unbalanced datasets. We compared four different groups of models: the draft GEMs reconstructed from CarveMe (CarveMe), gap-filled GEMs by adding the top 200 reactions predicted by CHESHIRE and NHP (CHESHIRE-200 and NHP-200), and gap-filled GEMs by randomly adding 200 reactions from the universal BiGG reaction pool (Random-200).

We observed a high variability across the draft CarveMe models to predict fermentation profiles. One model correctly predicts all phenotypes (F1 = 1.0) but 14 others fail to predict any (F1 = 0.0). Adding the 200 NHP-predicted reactions barely improves the phenotypic predictions and the improvement is even worse than that after randomly adding 200 reactions (Fig. 3b–e). To the contrary, CHESHIRE-200 increases the mean performances significantly (Fig. 3b–e; Supplementary Fig. 5a, b; $P < 0.01$, two-sided paired-sample $t$-test) and, in particular, the F1 score for 11 of the 24 draft GEMs. Compared across the 9 fermentation metabolites, the biggest improvement of CHESHIRE over CarveMe draft models was observed on acetic acid followed by lactic acid production. Among the 24 draft GEMs, CHESHIRE increases the correct predictions of acetic acid and lactic acid phenotypes from 8 to 17 and 14 to 20, respectively. Finally, we demonstrated that the improved performance is not simply due to more reactions by showing a significantly better performance of CHESHIRE-200 than that of Random-200 ($P < 0.05$, two-sided paired-sample $t$-test).

To test whether the improvement of CHESHIRE over CarveMe-reconstructed draft models is generalizable to GEMs from other sources, we performed the same fermentation test on ModelSEED-reconstructed draft models. Similar results were observed, where CHESHIRE-200 improves the predictions over draft models, and draft models plus randomly selected 200 reactions (Supplementary Fig. 6). NHP, again, shows no improvement. Our results suggest that CHESHIRE enables consistent improvement of phenotypic prediction over GEMs reconstructed from different pipelines.

The fermentation test comprises nine metabolites that are very close to central carbon pathways. To test if CHESHIRE can fill other types of gaps, we assessed CHESHIRE for predicting secretions of amino acids, substrate utilization for growth, and gene essentiality. The dataset of amino acid secretions measures production profiles of 20 amino acids for 25 bacterial GEMs (Supplementary Table 2, see "Amino acid secretion test data in Supplementary Note 3"). This dataset is highly unbalanced with 478 positive phenotypes and 22 negative phenotypes. Similar to the fermentation test described above, CHESHIRE-200 outperforms Random-200 (F1 score: $P < 10^{-5}$, two-sided paired-sample t-test), and NHP-200 shows no improvement at all (Fig. 3f–i; Supplementary Fig. 5c, d). In particular, CHESHIRE increases the correct predictions of 67 amino acid secretions that are knowledge gaps in draft GEMs. Despite the significant improvement, the recall remains very low at about 20%, suggesting many remaining false-negative gaps.

Each of the substrate utilization and gene essentiality tests contains 5 GEMs (Supplementary Table 3, see "Substrate utilization test data and Gene essentiality test data in Supplementary Note 3"). The utilization of various carbon-, nitrogen-, phosphorus-, and sulfur-substrates for growth were tested using Biolog phenotype arrays[37] in a high-throughput manner. Essential genes were identified using gene knockout experiments and a gene is essential if its deletion abolishes growth. When tested on both datasets, CHESHIRE, however, fails to fill gaps in nearly all 5 GEMs, except for the growth phenotypes of *Bucillus subtilis* where CHESHIRE-200 increases the F1 score of CarveMe draft model from 0.58 to 0.87 (Supplementary Fig. 7). Notably, NHP fails in all the tests, including the growth phenotypes of *B. subtilis*.

To understand how CHESHIRE gap-fills the GEMs, we used Mixed Integer Linear Programming to infer which reaction(s) from the top 200 candidates that causally fill the gaps (Fig. 3a, see "Causal reaction inference in Supplementary Note 5"). The gap-filling guided by CHESHIRE corrected 10 false-negative predictions by adding a single acetic acid transport reaction between intracellular and extracellular via proton symporter, e.g., *Prevotella bergensis* DSM 17361 (Fig. 3j). This is not surprising as the poor performance of draft GEMs is mostly due to poor annotation of transporter genes[38–40]. Beside false negatives, CHESHIRE can also identify false-positive predictions. For example, the draft GEM of *Anaerobutyricum hallii* has lactate dehydrogenase and can theoretically produce or use lactate for growth when lactate is present in the culture medium. The draft model predicts lactate production, since lactate utilization is dispensable for maximal biomass production. However, this prediction contradicts experimental data. CHESHIRE-200 fills this gap by adding two NAD(P)H-mediated redox reactions (Fig. 3k) that enable maximization of growth rate by consuming lactate. Supported by previous reports of lactate consumption[41,42], this example shows that CHESHIRE can identify missing reactions that have consequences on distant fermentation pathways via a global and systematic effect. For gaps in amino acid secretions, causal reaction inference indicates that these gaps were solved by adding amino acid transportation reactions (Fig. 3l, m).

To assess the false positives added by CHESHIRE-200, we counted the number of fermentation products and amino acids that were added by CHESHIRE but not observed experimentally. For the fermentation test, CHESHIRE adds a phenotype in 27 simulations (i.e., combinations of genome and metabolite), where 15 are true positives

and 12 are false positives. For the amino acid test, all 67 simulations that predicted a gain of phenotype are true positives. These results suggest that the risk of introducing false positive phenotypes may depend on the model and phenotype. We further assessed whether these false positives may be linked to the bias of CHESHIRE to score specific types of reactions higher than the others. For nearly all enzymatic functional classes (see "Enzymatic functional class of reactions in Supplementary Note 5"), we found a huge variability in the rankings of reactions catalyzed by enzymes that belong to each individual class (Supplementary Fig. 8). Relatively, reactions catalyzed by dinucleosidetriphosphatase, hydratase, and cyclase are scored higher on average.

## Discussion

Optimization-based GEM gap-filling has been long considered as a process of fitting a GEM to observed data[43]. This problem is typically formulated by a mixed-integer linear programming that minimizes the number of added reactions under the constraint that the observed phenotypes are satisfied. Therefore, the majority of GEM gap-filling methods falls short of predicting metabolic gaps in both network connections and functions without knowing experimental phenotypes a priori. FastGapFill, as one of a few exceptions, fits a specific task of gap-filling to resolve dead-ends and blocked reactions[21]. Previous studies[44–46] have shown that FastGapFill exhibits a poor performance in filling artificially introduced gaps. Although gap-filling with experimental data is critically important, it is limited to understanding the gene-reaction-phenotype mappings in conditions where the data was collected. The environmental conditions are combinatorially complex; as a theoretical tool, the primary purpose of GEMs is to rapidly offer theoretical predictions of metabolic activities over a large array of environmental conditions where data has not been collected.

We therefore present a method CHESHIRE which uses deep learning techniques to resolve gaps in GEMs at reaction and phenotypic levels solely based on network topology. It is completely unsupervised without any input phenotypic data and thus improper to be compared to data-driven optimization-based methods directly. The performance of CHESHIRE has been rigorously examined through both internal and external validations over GEMs of a large set of microorganisms. Compared to previous gap-filling methods, CHESHIRE adopts the concept of hypergraphs with advanced graph convolutional networks to accurately learn the geometrical patterns of metabolic networks and predict missing metabolic reactions without inputs from any experimental data. In addition, CHESHIRE is computationally efficient than C3MM and NHP (Supplementary Table 4, see "Complexity analysis in Supplementary Note 2"). Most importantly, CHESHIRE has been validated on realistic biological datasets. To our best knowledge, such benchmark has not been performed for previous topology-based gap-filling methods. We showed that CHESHIRE significantly improves the phenotypic predictions of fermentation products and amino acids secretions over a total of 49 draft GEMs reconstructed from a mostly used automatic reconstruction pipeline CarveMe[11]. Despite the success, we found that substrate utilization and gene essentiality are gap-filling resistant cases for CHESHIRE and, broadly, the topology-based gap-filling methods. Even in this worst scenario, CHESHIRE shows better performance than the competitive method NHP in the prediction of substrate utilization for *B. subtilis*. Since only 5 GEMs were used in each of the two tests, a comprehensive assessment of CHESHIRE over a larger GEM collection may be more informative of how much CHESHIRE struggles with these tasks.

Although CHESHIRE advances phenotypic predictions, the use of a universal pool and the top 200 reactions risk of adding reactions that do not exist (false positives). Correct predictions of phenotypes do not necessarily mean correct inference of missing reactions. It is likely that different enzymes carry the same metabolic functions (e.g., fermentation metabolism) from different substrates. While adding reactions is

expedited by increasingly advanced machine learning methods, it is still a heavily manual task to trim reactions that are wrongly added. This challenge, termed as content removal, has been recognized as one of the two fundamental bottlenecks for GEM quality improvement[47]. This bottleneck is largely overlooked and highly time-consuming: a significant amount of time is required to identify which reactions should be removed. Initial community-driven efforts have been made to mitigate this challenge, including building new GEM quality standards such as MEMOTE[48] and developing novel frameworks for GEM quality assessment and improvement.

In this study, we have taken initial steps towards further reduction of the number of CHESHIRE-predicted reactions. First, CHESHIRE-200 conducted a comprehensive search of 236 metabolites to identify metabolic phenotypes that can be potentially gap-filled by adding 200 reactions (see "Simulations of metabolic phenotypes in Supplementary Note 5"). The tool also provides information on the essential reactions for each potential gap, allowing users to focus on the most promising reactions without being overwhelmed by all the 200 reactions. Moreover, we explored the feasibility of prioritizing reactions with different cofactors (e.g., NADH) based on their prevalence in draft GEMs (see "Generation of GEMs in Supplementary Note 5"). We found that excluding candidate reactions involving less prevalent cofactors led to improved performance for the secretion of fermentation products, but decreased performance for the dataset of amino acid secretions (Supplementary Fig. 9). These contrasting trends suggest that the relationship between missing reactions and cofactors may depend on the secreted products and their biosynthetic pathways. Importantly, the co-factor-based strategy enabled CHESHIRE to reduce the number of added reactions to 100, while still significantly improving gap-filled GEMs over draft GEMs ($P < 0.01$, two-sided paired-sample $t$-test). Thus, our preliminary cofactor analysis highlights a promising future direction for metabolic network gap-filling.

Alternative to content removal, the number of false positives can be reduced by limiting the database size for candidate reactions. Though at its infancy, database reduction is a valuable technology that has broad utility for both optimization- and topology-based methods. There are many possible routes of database reduction. First, a universal database can be split into genus-, species-, or even phylogroup-specific databases by aggregating all reactions in GEMs that belong to individual taxa. Despite the pioneering efforts in AGORA models, we are still lacking a large-scale database of high-quality GEMs that cover a wide range of taxonomic diversity. Second, GapSeq[12] points out a promising direction to use genomic information for reducing a universal database to a small subset of reactions supported by gene annotations. By lowering the threshold for sequence homology, comparing protein domains and sequence signatures, and mapping content to distantly related organisms, more genes and candidate reactions can be annotated[47]. Finally, we should not ignore the possibility that gaps may be filled by altering the directionality of reactions without adding new ones. Neither the BiGG universal reaction database nor the current version of CHESHIRE considers reaction directionality. Therefore, CHESHIRE cannot fill gaps caused by wrong directions or reduce the number of added reactions by merging reactions with the same stoichiometry but different directionality. Further studies are warranted to incorporate reaction directionality in the CHESHIRE framework. A systematic integration of available thermodynamic data with GEMs will reduce the number of gaps and thus the total number of incorrectly introduced reactions. This deserves dedicated efforts, and we leave it as a future work.

## Methods

### Hypergraphs

As a natural extension of graphs, hypergraphs are composed of hyperlinks (also called hyperedges) which can join any number of nodes[25–28]. Hypergraphs are superior in modeling the correlation of practical data that could be far complex than pairwise patterns[49]. Mathematically, an unweighted hypergraph $\mathcal{H} = \{\mathcal{V}, \mathcal{E}\}$ where $\mathcal{V} = \{v_1, v_2, \ldots, v_n\}$ is the node set and $\mathcal{E} = \{e_1, e_2, \ldots, e_m\}$ is the hyperlink set with $e_p \subseteq \mathcal{V}$ for $p = 1, 2, \ldots, m$. Two nodes are called adjacent if they are in the same hyperlink. A hypergraph is called connected if given two nodes, there is a path connecting them through hyperlinks. An incidence matrix of a hypergraph, denoted by $\mathbf{H} \in \mathbb{R}^{n \times m}$, consists of logical values which indicate the relationship between nodes and hyperlinks. If a node $v_i$ is participated in a hyperlink $e_p$, then the $(i, p)$th entry of $\mathbf{H}$, i.e., $\mathbf{H}_{ip}$, has value one. If not, it is equal to zero.

### Feature initialization

In a transductive learning setting, the node attributes are not provided. It is thus necessary to generate the node features based on the hypergraph structure solely. Given an incomplete hypergraph $\mathcal{H}$ with $n$ nodes, we therefore propose an encoder-based approach to produce node features by simply passing the incidence matrix $\mathbf{H}$ through a one-layer neural network, i.e.,

$$\mathbf{x}_i = \text{hard-tanh}\,(\mathbf{W}_{\text{enc}}\mathbf{h}_i + \mathbf{b}_{\text{enc}}) \text{ for } i = 1, 2, \ldots, n, \tag{1}$$

where $\mathbf{h}_i$ denotes the $i$th row of the incidence matrix, $\mathbf{W}_{\text{enc}}$ and $\mathbf{b}_{\text{enc}}$ are the learnable parameters in the encoder, and hard-tanh is a nonlinear activation function defined as

$$\text{hard-tanh}\,(x) = \begin{cases} 1 & \text{if } x \geq 1 \\ -1 & \text{if } x \leq -1 \\ x & \text{otherwise} \end{cases}. \tag{2}$$

Hard-tanh is more efficient to compute while maintaining or improving the performance of deep neural networks (compared to tanh)[50]. Incidence matrix of a hypergraph is able to capture multidimensional relationships unambiguously while keeping low memory costs[51]. Hence, we believe that our approach can provide more accurate initial node features of a hypergraph with less computational costs.

### Feature refinement

Feature refinement is the most critical component in CHESHIRE, which is composed of normalization, dropout, and graph convolutional networks. First, we decompose the hypergraph into a disjoint graph with separate cliques formed by the hyperlinks. Two nodes in the disjoint graph share the same feature space if originating from the same node in the hypergraph. After obtaining the node features of the disjoint graph, we feed the features to a graph normalization layer for each clique. Suppose that the dimension of the feature vectors is $d_{\text{enc}}$. Let $\mathbf{x}_{ij}$ denote the $j$th entry of the feature vector $\mathbf{x}_i$ for node $v_i$. Then the element-wise normalized features are given by

$$\tilde{\mathbf{x}}_{ij} = \gamma_j \frac{\mathbf{x}_{ij} - \alpha_j \mu_j}{\sigma_j} + \beta_j \text{ for } j = 1, 2, \ldots, d_{\text{enc}}, \tag{3}$$

where $\alpha_j$ is a learnable parameter that controls how much information need to keep in the mean, $\gamma_j$ and $\beta_j$ are the affine parameters, and $\mu_j$ and $\sigma_j$ are the mean and standard derivation of the features in each clique, respectively. Graph normalization has proved to be advantageous in training graph convolutional networks compared to other normalization methods such as batch and layer normalization[52]. In order to prevent overfitting, we further add an alpha dropout layer after the graph normalization. The alpha dropout utilizes a scaled exponential linear unit (SELU), which includes self-normalizing properties such as maintaining the mean and standard derivation of the inputs and avoiding exploding and vanishing gradients[53]. For convenience, we drop the tilde notation and use $\mathbf{x}_i$ as the updated features.

Second, we continue to refine the features with a CSGCN on each clique (corresponding to a hyperlink in the original hypergraph).

CSGCN exploits the Chebyshev polynomial expansion and spectral graph theory to learn the localized spectral filters which can extract local and composite features on graphs that encode complex geometric structures[32]. Given a hyperlink $e_p \in \mathcal{E}$, we refine the features of node $v_i$ by

$$\hat{\mathbf{x}}_i = \text{hard-tanh}\left(\sum_{k=1}^{K} \mathbf{W}_{\text{conv}}^{(k)} \mathbf{z}_i^{(k)}\right) \text{ for } v_i \in e_p, \quad (4)$$

where $K$ is the Chebyshev filter size, $\mathbf{W}_{\text{conv}}^{(k)}$ are the learnable parameters in the CSGCN, and $\mathbf{z}_i^{(k)}$ are computed recursively as

$$\mathbf{z}_i^{(1)} = \mathbf{x}_i, \quad \mathbf{z}_i^{(2)} = \tilde{\mathbf{L}}\mathbf{x}_i, \quad \text{and} \quad \mathbf{z}_i^{(k)} = 2\tilde{\mathbf{L}}\mathbf{z}_i^{(k-1)} - \mathbf{z}_i^{(k-2)}. \quad (5)$$

The matrix $\tilde{\mathbf{L}}$ is the scaled normalized Laplacian matrix defined as

$$\tilde{\mathbf{L}} = \frac{2}{\lambda_{\max}}\mathbf{L} - \mathbf{I} = \frac{2}{\lambda_{\max}}(\mathbf{I} - \mathbf{D}^{-\frac{1}{2}}\mathbf{A}\mathbf{D}^{-\frac{1}{2}}) - \mathbf{I}, \quad (6)$$

where $\mathbf{L}$ is the symmetric normalized Laplacian matrix of the clique with the largest eigenvalue $\lambda_{\max}$, and $\mathbf{D}$ and $\mathbf{A}$ are the degree and adjacency matrices of the clique, respectively. For convenience, we drop the head notation and use $\mathbf{x}_i$ as the updated features.

## Pooling and scoring
We aggregate the refined node features within each clique/hyperlink to produce a score. There are many pooling functions such as mean pooling and maximum pooling. Here we use two different pooling functions. Suppose that the dimension of the convolutional feature vector is $d_{\text{conv}}$ and denote $\mathbf{x}_{ij}$ as the $j$th entry of $\mathbf{x}_i$. We propose to employ a Frobenius norm-based (also known as the $l_2$-norm) pooling function to generate hyperlink features, which is defined as

$$\left(\mathbf{y}_p^{(\text{norm})}\right)_j = \left(\frac{1}{|e_p|}\sum_{v_i \in e_p}\mathbf{x}_{ij}^2\right)^{\frac{1}{2}} \text{ for } j = 1, 2, \ldots, d_{\text{conv}}. \quad (7)$$

Norm-based pooling functions are more efficient at representing complex and nonlinear separating boundaries and has been widely used in traditional convolutional neural networks[33].

In order to achieve a better performance, we also incorporate the maximum minimum-based pooling function[22] defined as

$$\left(\mathbf{y}_p^{(\text{maxmin})}\right)_j = \max_{v_i \in e_p}\{\mathbf{x}_{ij}\} - \min_{v_i \in e_p}\{\mathbf{x}_{ij}\} \text{ for } j = 1, 2, \ldots, d_{\text{conv}}. \quad (8)$$

Therefore, the final score of a hyperlink $e_p$ is then given by

$$S_p = \text{sigmoid}\left(\mathbf{W}_{\text{score}}(\mathbf{y}_p^{(\text{maxmin})}||\mathbf{y}_p^{(\text{norm})}) + \mathbf{b}_{\text{score}}\right), \quad (9)$$

where "||" denotes the vector concatenation operation, and $\mathbf{W}_{\text{score}}$ and $\mathbf{b}_{\text{score}}$ are the learnable parameters in the scoring neural network. Empirically, we found that the combination of the two pooling functions can make full use of their own advantages, leading to a better performance during internal validation.

## Training algorithm
We train CHESHIRE with the following loss function

$$\text{Loss} = \frac{1}{|\mathcal{E}|}\sum_{e \in \mathcal{E}}\sigma\left(\left(\frac{1}{|\mathcal{F}|}\sum_{f \in \mathcal{F}}S_f\right) - S_e\right), \quad (10)$$

where $\mathcal{E}$ is the set of positive hyperlinks, $\mathcal{F}$ is the set of negative hyperlinks, and $\sigma(\cdot) = \log(1 + \exp(\cdot))$ is the logistic function[22,54]. We

chose the above loss function since it offers a better performance compared to traditional classification loss functions such as cross entropy loss. We exploit the highly efficient Adam optimization algorithm to train CHESHIRE. During the training stage, CHESHIRE tries to learn the weights of the deep neural network by minimizing the loss function, which maximizes the scores for positive hyperlinks to be higher than the average score for negative hyperlinks. During the testing stage, CHESHIRE uses the learned weights to calculate a probability score for an unseen hyperlink (from either a testing set or a universal set).

## Hyperparameter selection
The key hyperparameters of CHESHIRE are the encoder feature dimension, the graph convolutional feature dimension, the Chebyshev filter size, the dropout probability, and the learning rate. We used a universal hyperparameter set for CHESHIRE during internal and external validations. We found that the performance of CHESHIRE with a pre-selected universal hyperparameter set is close to that obtained by grid search. This implies that CHESHIRE is not sensitive to these hyperparameters. Therefore, we decided to use a universal hyperparameter set for all the GEMs, which can also save a great amount of computational resources. The encoder feature dimension, the graph convolutional feature dimension, the Chebyshev filter size, the dropout probability, and the learning rate are set to 256, 128, 3, 0.1, and 0.01, respectively. For hyperparameters of NVM, C3MM, and NHP, see "Hyperparameter selection in Supplementary Note 4".

## Negative sampling
In order to accurately predict missing reactions from a metabolic network, it is necessary to sample negative reactions, i.e., reactions that do not exist. We used the negative sampling strategy proposed in[22]. Suppose that we have a hypergraph $\mathcal{H} = \{\mathcal{V}, \mathcal{E}\}$ that captures a metabolic network. For each (positive) hyperlink $e \in \mathcal{E}$, we generate a corresponding negative hyperlink $f$, where half of the nodes in $f$ are from $e$ (rounding is required for odd number of metabolites) and the remaining half are from $\mathcal{V} - e$ (the set of nodes that are not in $e$). The motivation behind the strategy is that it is extremely unlikely that half of the metabolites from a valid reaction and randomly sampled metabolites are participated in another valid reaction. We denote the set of negative hyperlinks as $\mathcal{F}$, in which the number of negative hyperlinks are equal to the number of positive hyperlinks.

## Reporting summary
Further information on research design is available in the Nature Portfolio Reporting Summary linked to this article.

## Data availability
This study used publicly available data from publications of the scientific literature and accessible repositories (see "Supplementary Note 3"). Source data are provided with this paper.

## Code availability
The source code for our computational framework is available at Github[55] [https://github.com/canc1993/cheshire-gapfilling].

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

## Acknowledgements

This work is supported by the the National Institutes of Health (R01AI141529, R01HD093761, RF1AG067744, UH3OD023268, U19AI095219, and U01HL089856) to Y.Y.L.

## Author contributions

Y.-Y.L. conceived and designed the project. C.C. developed the CHE-SHIRE algorithm and performed the internal validation. C.C. and C.L. performed the external validation. C.L. interpreted the results. C.C. and C.L. prepared the manuscript. Y.-Y.L. edited and approved the manuscript.

## Competing interests

The authors declare no competing interests.
