## [Peer Review File · Nature Communications]

Teasing out Missing Reactions in Genome-scale Metabolic Networks through Hypergraph LearningReviewers' Comments:

Reviewer #1:

Remarks to the Author:

General:

The first sequenced microbial genomes, appearing in the mid to late 1990s, led to the reconstruction of genome-scale metabolic networks that in turn enabled the formulation of the first genome-scale models (GEMs) of metabolism. GEMs brought into focus the incompleteness of genome annotation and the knowledge that we have available on gene function. Gaps in knowledge were reflected in gaps in reconstruction metabolic networks. Thus, in the mid 2000, a series of algorithms were developed to help fill in these knowledge gaps. This became known as gap-filling, that is the subject matter of this paper.

The rapid drop in the cost of DNA sequencing in the late 2000s, lead to an exponentially growing number of whole genome sequences for bacteria. By the late 2010s this led to the availability of 10s of thousands of high-quality genome sequences. The need to reconstruct metabolism thus grew massively. An obstacle to formulating GEMs across the bacterial phylogenetic tree is the lack of availability of high-quality gap-filling algorithms that lead to robust metabolic models. This is likely to remain a growing challenge throughout the 2020s. This is thus a timely study that addresses a key issue

This paper aims to introduce a novel ML-based gap-filling approach to address current challenges with gap-filling of metabolic networks in a scalable and faster fashion. This paper contains a wealth of information, New deep-learning based methodology introduced by authors has the advantage of preserving higher order structural information, and efficient training process, this results in a scalable and faster gap-filling solution compared to other state of the art ML-based gap-filling approaches such as NPH. Conventional constraint-based gapfilling approaches (Gapfill-Gapfind, Fast-gapfill etc,) are far from being perfect due to scalability problems and computation time. In these approaches, gap-filling of metabolic networks is frequently done by adding a minimum number of reactions from a reference database that facilitate growth under a chemically defined medium, due to this limitation, since many microorganisms are not culturable and subsequently there is no formulated minimal defined media available for them, constraint-based gap-filling approaches haven't been applicable to such organisms. Also, there are many reports on non-feasible gap-filling solution for newly reconstructed models using current approaches due to inefficient and time-consuming search in polygonal solution space. All in all, new methodology introduced by authors overcomes many problems with current gap-filling approaches, but I do have some fundamental concerns about what authors claimed here.

General comments:

1) the paper is very compsci/bioinformatics in nature. Will be inaccessible to the typical life scientist as now written

2) background literature and contextualization of results well done. There is a recent paper in CELL SYSTEMS on the 'Life cycle of GEMs' that might have some material that would be appropriate for the discussion. There is also community-driven development of quality standards for GEMs called MeMote, that appeared in Nature Biotech recently that would be a relevant reference that the authors should consider for the Discussion section.

3) I cannot check this myself, but prompt dissemination of the code would be highly important for widespread adoption.

4) Validation is a key issue for this paper. 'Internal' validation is done by introducing 'artificial gaps' in existing curated models and CHESHIRE is compared to other well-known gap filling tools. This is a

modestly stringent text as these models are well curated. They authors show statistically that CHESHIRE performs better than three existing tools.

5) Detailing the advantages and disadvantages of CHESHIRE would be helpful. A biological/biochemical assessment of what kinds of gaps does CHESHIRE fill in easily, and which does it struggle with.

6) Another test bed would be gap-filling in a pangenomic setting. I.e., starting with 100s or even 1000s of available whole genome sequences of the same species (or even a phylogroup within a species) would help with comparative analysis between closely related genomes, and the mass of data from multiple strains might help gap fill across a species (or phylogroup). This would be a notable advance, given the forecast of genome availability on the 2020s

7) there are cases of 'gap-filling resistant' cases, where none of the current tools can fill a gap. If CHESHIRE can gap fill such cases, it is not only statistically better, but also better 'in kind' compared to alternative methods.

Technical comments:

1) Considering the large number of deposited genomes on public databases and current progress on pangenome analysis there is an emerging trend of generating large-scale metabolic reconstruction which demands scalable and fast gap-filling approaches. Although authors claimed that CHESHIRE is faster and more scalable than the current ML-based approaches, data on computation time and applying the CHESHIRE on a large set of draft networks are not provided in this research.

2) In some cases, gaps are produced by reactions directionality rather than lack of a specific reaction, it seems Hypergraph does not consider reactions directionality. Which might affect the number of added reactions during gap-filling.

3) Although BiGG universal reactome contains a curated pool of reactions, it does not consider reactions directionality, also it contains same metabolite and reactions with different identifiers. In this regard, did authors curate these inconsistencies before using BiGG universal reactome? If not, how do they justify using an inconsistent pool of reactions for validation of their work?

4) For external validation, Authors claimed that by adding the top 200 or 500 reactions with high confidence score will result in a gapfilled model with more precise secreted-metabolites prediction, considering the size of a normal bacterial GEM (800-200 reactions), it doesn't make sense to add 200/500 reactions during gapfilling, unless there would be more than 500 genes missing from genome annotations, which is unrealistic.

5) In regard with external validation, authors only used draft models generated by CarveMe pipeline, though if their methodology is strong enough to fill metabolic gaps efficiently, it should be generalizable to the GEMs generated by other pipelines (Modelseed, Gapseq, Merlin, etc.), it is highly recommended to diversify testing set not only by randomization of reactions deletion but also by considering GEMs from different sources.

6) Both internal and external validations are performed on high quality data sets. BiGG models are known as most curated highly consistent models which mostly generated on model organisms with less knowledge gaps. Also, external validation is uniform, in this case, models are generated from high quality reference genomes which subsequently will not be problematic during gap-filling due to a smaller number of missing genes from annotation. Also, all of them have been generated using the same pipeline, which will affect the results in a biased manner.

7) Title of the article should be more detailed, the deep learning is a general terminology, it would be

more precise if authors consider reforming title based on specific approach they've used (graph convolutional networks).

8) It is recommended that authors consider a graphical abstract illustration which explains network architecture in detail, it would help readers skim through the provided workflow easier.

9) Judging gapfilling efficiency by capability of fermentation profile is not a strong argument toward defending what authors have claimed in this research, it is known that different enzymes could produce same fermentation metabolism from different substrates. Therefore adding reaction from a universal reaction pool (when reaction pool is not strain/species or even genus specific reaction pool) might result in right phenotype but with wrong reactions. For example, false-positive phenotype of *Anaerobutyricum hallii* draft network (15) might be simply solved by checking the directionality of reaction instead of adding 500 reactions to make it predict better. Or in worst case scenario adding an isozyme of lactate dehydrogenase with different co-factor might also be helpful for solving such a simple problem

Reviewer #2:

Remarks to the Author:

The manuscript presents a deep learning-based method — CHEbyshev Spectral HyperInk pREdictor (CHESHIRE) — to predict missing reactions of genome-scale metabolic models (GEMs) purely from the metabolic network topology. The authors claim that the method outperforms other topology-based methods in 108 high-quality GEMs. Actually, the authors correctly state in the abstract that they offer "compelling evidence that CHESHIRE outperforms other topology-based methods," as this kind of method can only be compared by simulation experiments, and the results of the comparison still depend strongly on the tested conditions.

The manuscript does a perfect job of describing the problem, the current procedures, and the challenges of the gap-filling process.

The gap-filling process involves adding reactions in a metabolic model to have the model consistent with observed phenotypes, such as growth assays (on various substrates), metabolite production, and true-positive and true-negative predictions in knock-out experiments.

In general, there are two main approaches to introducing reactions that improve model consistency: (1) optimization-based approaches that seek to maximize model consistency with observed phenotypes (production or consumption of specific metabolites, growth, etc.) by introducing, from a given database, a min number of reactions (although one can identify multiple alternative minimal or great sets of reactions), and (2) network topology methods of which, the latest and improved methods use ML methodologies. Both of these approaches use precisely the same starting inputs: an initial draft model and a reference database of reactions used for gap-fill.

The network topology methods first introduce many reactions, depending on the technique and user-defined parameters. They then use a set of phenotypic data to test if the gap-filled model can simulate these phenotypes. However, there are still many reactions that are now part of the model for which there is no genetic or physiological justification for their presence.

In the optimization-based approach, the phenotypic data are used in advance. The methods guarantee that the reactions added to the model are necessary (even as an alternative) for simulating the observed phenotype. In this process, no reaction is added, which does not contribute to the observed phenotype.

In general, both methods could provide the result, the computational effort could be comparable, and they both involve a manual curation process, as the last process in the topology-based approach is a manual curation.

If the authors agree with the above statements, then we would agree that some of their statements

about the lack of manual curation and the no need for phenotypic data in ML network-topology-based methods are strong, and they can be misleading.

Another major issue appears to exist in the comparison between the optimization-based methods and the CHESHIRE method (and the topology-based methods), the conclusions, and the statements about the results of the comparison.

It is unclear if the authors performed the CarveMe reconstruction themselves or if they used the models in the CarveMe database. Even if the authors used the CarveMe workflow and reconstructed the so-called draft models, did they use only the growth as a phenotype for gap-fill within the CarveMe workflow, or did they use the fermentation data to obtain the draft? I assume that the authors used only the growth as a phenotype for gap-fill, which is the default procedure for the published methodology of CarveMe. In this case, I would argue that the CarveMe draft models did not "know the experimental phenotypes a priori." In this respect, their performance is very good since no reaction is added except for a minimal set for growth. To perform a fair comparison, the authors should compare the models from their method with models derived using the optimization gap-fill method in CarveMe, which uses the fermentation data as input in advance, since the author claim that their method does better than the optimization-based methods that use the phenotypic data as a priori knowledge.

It is also unclear if the authors used the same (extracellular) media as in the CarveMe protocol. The definition of media is crucial for model reconstruction and validation, and the topology methods do not consider it. The authors here do not clearly define it and justify the one they use.

Finally, I think that some major issues reduce the significance and broader impact of the manuscript.

First, the improvement of the method relative to other ML, network-topology-based methods is insignificant and only provided as evidence. A larger set of models and phenotypes should be used to test that at least improvement is consistent and quantitatively better.

Second, the models derived from the CHESHIRE method have an order of 500 reactions which might have no physiological relevance. I am afraid that the community and the users of these models would like to have some justification for the excessive (useless) number of reactions. How could the authors trim these reactions without going through a manual curation? Unless they suggest a manual curation, which goes against the main claimed advantage of their method (no need for manual curation). Although, as we discussed above, the network-topology methods still involve much effort in manual curation.

Third, the phenotypic data comprises nine metabolites that are very close to central carbon pathways. The methods should be tested for various produced metabolites (or phenotypes that involve metabolites) distributed across different classes of metabolites (amino acids, lipids, secondary metabolites, terpenoids, etc.). These compounds are in complex pathways, and the gap-filling of these pathways is a very challenging problem for any gap-filling method.

Finally, the models derived from the CHESHIRE, and similar methods, should be tested for their performance against growth assays and knock-out experiments, using the metrics used in these studies and model testing and validation (such as the overall accuracy and the Matthews Correlation Coefficient). These essential metrics will also evaluate the number of (mainly) false-positive predictions due to the large number of reactions introduced in the network.

The final suggestion in the discussion, about the use of CHESHIRE and similar methods for database reduction also used from the optimization methods, is very valuable and could have a broader impact. But it will require a specific and well-designed investigation.

Reviewer #3:

Remarks to the Author:

Chen, Liao and Liu present a method to predict missing reactions in genome-scale metabolic networks (GEMs) through deep learning based on CHEbyshev Spectral Hyperlink pREdictor (CHESHIRE). This method is based on network topology: given a metabolic network represented as metabolite-reaction incidence matrix, it learns the network structure, and then for a given reaction provides a score indicating how likely this reaction is part of the input network. The authors demonstrate the better performance of their method compared to the other deep learning-based and topology-based gap filling methods on an „internal“ validation dataset, where the removed reactions are known, and an „external“ validation dataset, where the missing reactions are not known. For the latter, the performance metric is the predicted fermentation phenotype of 9 metabolites, which is compared to experimentally determined phenotypes. The authors also provide biological interpretations of the gap-filled reactions that help to recapitulate the experimentally determined phenotype.

In general, the method is promising, and the application of deep learning in genome-scale metabolic modelling will be interesting for a broader audience interested in machine learning applications in biology, mathematical modelers, and system biologists. However, there is a number of issues that need to be clarified or addressed in order to assess the utility of this method for gap filling of metabolic networks in biological applications.

Major comments:

1) From Figure 1d,e and model architecture and training description on pages 7 and 8 it does not become fully clear how the model is trained and what exactly the input data is. On page 5, it is written that CHESHIRE takes a metabolic network and a pool of candidate reactions as input, and produces confidence scores for the candidate reactions as output. However, the input on Figure 1d only depicts metabolic network, and misses the candidate reactions. On page 7, it is written that the pooling step computes reaction features, does this mean for reactions from the incidence matrix? How will the reaction features be computed for the pool of candidate reactions? Is incidence matrix also created for them?

2) Importantly, what is the loss function that is optimized during the training step? On page 8, it is written that the input model is enhanced with negative reactions („fake“ reactions added in the same amount as real reactions). Are these negative reactions also included as input, and is the model provided with the information on which reactions are true, and which are false? It seems like this is a crucial step in model training, but it is not described neither in Figure 1e nor explicitly in the text.

3) On page 9, it is written that positive and negative reactions from each model are split into 60% training (metabolic network to be gap-filled) and 40% testing (unseen candidate reactions). Has these procedure been repeated multiple times for different splits in 60% and 40%? And what about validation, how were the model parameters tuned during the training phase? Was subset of the 60% used for optimization of model parameters?

4) On page 8, it is written that negative reactions are not mandatory for algorithm testing. But in this section, reactions from the model (40%) are provided for testing, so if only positive reactions will be in the candidate pool, what is the point of assessing performance without negative reactions?

5) For internal validation, providing 40% of model reactions and artificially created negative counterparts as candidate reaction pool seems to be over-optimistic, since in real application the pool of candidate reactions will be much bigger and much less balanced in terms of positive and negative classes (as is the case with external validation). Since all the models used for internal training are curated BiGG models, it would be interesting to see how CHESHIRE and other methods perform if the candidate reaction pool is not coming from the same model with some negative reactions added, but

agnostically use BiGG database pool as for external validation, and demonstrate whether CHESHIRE is able to identify original reactions.

6) For external validation, the best results were achieved when 500 reactions were added with CHESHIRE, but 500 gap-filled reactions are quite a lot given that models usually have 1500-2000 reactions. Manual gap filling processes are usually adding much fewer reactions (10-100). How did the author assess false positives added in CHESHIRE-500? E.g., how many fermentation products were added, which were not observed experimentally? Is CHESHIRE biased towards scoring specific types of reactions higher than others (e.g. transport reactions, reactions involving cofactors, etc?) It would be interesting to analyze what types of reactions are scored highest, and whether there is statistical overrepresentation of certain types (e.g. with procedures similar to pathway enrichment analysis).

Minor comments:

7) On page 5, it is written that the main limitation of C3MM is that it cannot predict unseen reactions, since it includes all candidate reactions obtained from a universal reaction pool. However, it seems like CHESHIRE has exactly the same limitation, as it requires candidate reactions as input and provides their scoring. This limitation should be explicitly stated and discussed.

8) Page 15, it is written that „lactate consumption is preferred over production as it increases maximum growth rate“. How is growth rate included into the CHESHIRE pipeline, so that the authors can make this suggestion? Are the reports that the bacteria under study (*Anaerobutyricum hallii*) can consume lactate?

9) Figure 1, CHESHIRE schematic should include details about input candidate reaction pools and whether positive/negative labels are provided as well.

10) Page 28, legend to Figure 1 g: „We identified gaps of intact draft GEMs by comparing model predictions with phenotypic data“ – this sentence should be rephrased for clarity. I guess what is meant that the CHESHIRE predictions of filled gaps were assessed by comparing the gap-filled model performance to the original model in terms of fermentation reactions (phenotypic data), or something like that.

11) Figure 3 b-i, why are performance metric separated for NHP and CHESHIRE? It seems like all the methods could be put together on the four plots (AUROC, Recall, Precision, F1 Score).

12) Page 14, it is written that mean performance significantly increase. If the word „significant“ is used, the p-value and the name of the statistical test should be provided as well.

13) Typos:

- a. Page 1 Abstract: „spctral“
- b. Page 4 „involving -> involved“
- c. Page 5 „scalibility“

Response to Reviewer #1

Point 1.0. General:

The first sequenced microbial genomes, appearing in the mid to late 1990s, led to the reconstruction of genome-scale metabolic networks that in turn enabled the formulation of the first genome-scale models (GEMs) of metabolism. GEMs brought into focus the incompleteness of genome annotation and the knowledge that we have available on gene function. Gaps in knowledge were reflected in gaps in reconstruction metabolic networks. Thus, in the mid 2000, a series of algorithms were developed to help fill in these knowledge gaps. This became known as gap-filling, that is the subject matter of this paper.

The rapid drop in the cost of DNA sequencing in the late 2000s, lead to an exponentially growing number of whole genome sequences for bacteria. By the late 2010s this led to the availability of 10s of thousands of high-quality genome sequences. The need to reconstruct metabolism thus grew massively. An obstacle to formulating GEMs across the bacterial phylogenetic tree is the lack of availability of high-quality gap-filling algorithms that lead to robust metabolic models. This is likely to remain a growing challenge throughout the 2020s. This is thus a timely study that addresses a key issue

This paper aims to introduce a novel ML-based gap-filling approach to address current challenges with gap-filling of metabolic networks in a scalable and faster fashion. This paper contains a wealth of information, New deep-learning based methodology introduced by authors has the advantage of preserving higher order structural information, and efficient training process, this results in a scalable and faster gap-filling solution compared to other state of the art ML-based gap-filling approaches such as NPH. Conventional constraint-based gapfilling approaches (Gapfill-Gapfind, Fast-gapfill etc) are far from being perfect due to scalability problems and computation time. In these approaches, gap-filling of metabolic networks is frequently done by adding a minimum number of reactions from a reference database that facilitate growth under a chemically defined medium, due to this limitation, since many microorganisms are not culturable and subsequently there is no formulated minimal defined media available for them, constraint-based gap-filling approaches haven't been applicable to such organisms. Also, there are many reports on non-feasible gap-filling solution for newly reconstructed models using current approaches due to inefficient and time-consuming search in polygonal solution space. All in all, new methodology introduced by authors overcomes many problems with current gap-filling approaches, but I do have some fundamental concerns about what authors claimed here.

Response: We thank Reviewer #1 for reviewing our manuscript and thoroughly summarizing our work. Next, we address each of her/his comments in order.

Point 1.1. General Comments:

1) The paper is very compsci/bioinformatics in nature. Will be inaccessible to the typical life scientist as now written.

Response: We thank Reviewer #1 for this critical comment. In the revised manuscript, we have removed jargons unless they are absolutely needed for accurate description of the computational techniques. We also moved a substantial amount of technical descriptions to Supplementary Information to improve the readability of the main text. We hope that our revised version is well-received by a broad audience, including computational biologists, systems biologists, life scientists, engineers, and computer scientists.

Point 1.2. 2) background literature and contextualization of results well done. There is a recent paper in CELL SYSTEMS on the 'Life cycle of GEMs' that might have some material that would be appropriate for the discussion. There is also community-driven development of quality standards for GEMs called MeMote, that appeared in Nature Biotech recently that would be a relevant reference that the authors should consider for the Discussion section.

Response: We thank Reviewer #1 for suggesting the two important papers. Both "GEM life cycle" and "MeMote" suggest paths to standardize the production of high-quality GEMs, where gap-filling is an important step in the stage of model maturation. In the revised manuscript, we have cited both papers and discussed how CHESHIRE would benefit from the ongoing GEM quality assessment and improvement efforts. We have updated the corresponding text in the revised manuscript (see Page 17, Lines 312-323):

"Although CHESHIRE advances phenotypic predictions, the use of a universal pool and the top 200 reactions risk of adding reactions that do not exist (false positives). Correct predictions of phenotypes

do not necessarily mean correct inference of missing reactions. It is likely that different enzymes carry the same metabolic function (e.g., fermentation metabolism) from different substrates. While adding reactions is expedited by increasingly advanced machine learning methods, it is still a heavily manual task to trim reactions that are wrongly added. This challenge, termed as content removal, has been recognized as one of the two fundamental bottlenecks for GEM quality improvement [47]. This bottleneck is largely overlooked and highly time-consuming: a significant amount of time is required to identify which reactions should be removed. Initial community-driven efforts have been made to mitigate this challenge, including building new GEM quality standards such as MEMOTE [48] and developing novel frameworks for GEM quality assessment and improvement.”

Point 1.3. 3) I cannot check this myself, but prompt dissemination of the code would be highly important for widespread adoption.

Response: We thank Reviewer #1 for pointing this out. We have included the GitHub page for CHESHIRE in the Code Availability statement (see Page 18, Lines 347-348):

“Code Availability: The source code of our computational framework is available at <https://github.com/canc1993/cheshire-gapfilling>.”

Point 1.4. 4) Validation is a key issue for this paper. ‘Internal’ validation is done by introducing ‘artificial gaps’ in existing curated models and CHESHIRE is compared to other well-known gap filling tools. This is a modestly stringent text as these models are well curated. They authors show statistically that CHESHIRE performs better than three existing tools.

Response: We thank Reviewer #1 for this valuable comment. In the revised manuscript, we have performed the same internal validation (first type) on not-well-curated models, i.e., the 818 AGORA models of gut bacteria [1]. The AGORA models were built by first automatically generating draft reconstructions using ModelSEED, and then semi-automatically curated to improve the quality of the reconstructions. They are generally considered as of intermediate quality – between high-quality BiGG models and low-quality draft reconstructions. Using the AGORA models, we observed a consistent improvement of CHESHIRE over the other gap-filling methods such as NHP and C3MM (see Fig. R1, corresponding to Fig. S4 in the revised Supplementary Information). Therefore, our new results indicate that the performance of CHESHIRE is robust and insensitive to the input GEM quality. We have updated the corresponding text in the revised manuscript (see Page 9, Lines 159-161):

“We also performed the same type of internal validation on a larger set of metabolic networks, i.e., 818 AGORA models from the Virtual Metabolic Human database [8], and observed similar results (Fig. S4).”

Point 1.5. 5) Detailing the advantages and disadvantages of CHESHIRE would be helpful. A biological/biochemical assessment of what kinds of gaps does CHESHIRE fill in easily, and which does it struggle with.

Response: We thank Reviewer #1 for this critical suggestion. To find out the types of gaps that can be easily filled by CHESHIRE, we expanded our validation to allow for more kinds of gaps in metabolic functions. In addition to fermentation products that were tested in the previous submission, we have added new assessments during the revision to test the performance of CHESHIRE in filling other gaps in amino acid secretion, substrate utilization, and gene essentiality. We found that CHESHIRE improves the predictions of fermentation products (24 GEMs) and amino acids (25 GEMs) over intact draft models, draft models plus randomly added reactions, and draft models plus NHP-predicted reactions (see Fig. R9, corresponding to Fig. 3 in the revised manuscript). We have updated the corresponding text in the revised manuscript (see Page 13, Lines 230-240):

“The fermentation test comprises nine metabolites that are very close to central carbon pathways. To test if CHESHIRE can fill other types of gaps, we assessed CHESHIRE for predicting secretions of

amino acids, substrate utilization for growth, and gene essentiality. The dataset of amino acid secretions measures production profiles of 20 amino acids for 25 bacterial GEMs (Table S2, Supplementary Information Section 4.5). This dataset is highly unbalanced with 478 positive phenotypes and 22 negative phenotypes. Similar to the fermentation test described above, CHESHIRE-200 outperforms Random-200 (F1 score: $P < 10^{-5}$, two-sided paired-sample t-test), and NHP-200 shows no improvement at all (Fig. 3f-i, Fig. S5). In particular, CHESHIRE increases the correct predictions of 67 amino acid secretions that are knowledge gaps in draft GEMs. Despite the significant improvement, the recall remains very low at about 20%, suggesting many remaining false-negative gaps.”

When tested on substrate utilization and gene essentiality, both CHESHIRE and NHP, however, fail to fill the gaps present in nearly all GEMs (see Fig. R5), suggesting a shared limitation of topology-based gap-filling methods. The only exception is the GEM of *Bacillus subtilis*, for which CHESHIRE increases the F1 score from 0.58 (intact draft models) to 0.87 (draft models plus 200 CHESHIRE-predicted reactions). Since only 5 GEMs were used in each of the two tests, a comprehensive assessment of CHESHIRE over a larger GEM collection may be more informative of how much CHESHIRE struggles with growth phenotype and gene essentiality. We have updated the corresponding text in the revised manuscript (see Page 13-14, Lines 241-248):

*“Each of the substrate utilization and gene essentiality tests contains 5 GEMs (Table S3, Supplementary Information Section 4.6 and 4.7). The utilization of various carbon-, nitrogen-, phosphorus-, and sulfur-substrates for growth were tested using Biolog phenotype arrays [38] in a high-throughput manner. Essential genes were identified using gene knockout experiments and a gene is essential if its deletion is lethal and causes cell death. When tested on both datasets, CHESHIRE, however, fails to fill gaps in nearly all 5 GEMs, except for the growth phenotypes of *Bacillus subtilis* where CHESHIRE-200 increases the F1 score of CarveMe draft model from 0.58 to 0.87 (Fig. S7). Notably, NHP fails in all the tests, including the growth phenotypes of *B. subtilis*.”*

Point 1.6. 6) Another test bed would be gap-filling in a pangenomic setting, i.e., starting with 100s or even 1000s of available whole genome sequences of the same species (or even a phylogroup within a species) would help with comparative analysis between closely related genomes, and the mass of data from multiple strains might help gap fill across a species (or phylogroup). This would be a notable advance, given the forecast of genome availability on the 2020s

Response: We thank Reviewer #1 for raising this interesting point. We believe that gap-filling genomes of a specific species will greatly benefit from such a pangenomic setting when an increased number of genomes of this species can be well annotated. If genomes are poorly annotated, solely increasing their availability and using their draft GEMs would do little help on gap-filling: genes with unknown functions remain as knowledge gaps irrespective of the number of genomes. Currently, we are still lacking large-scale collections (100s or even 1000s) of high-quality GEMs that allow us to do such pangenomic gap-filling even for the model organisms such as *Escherichia coli*. Other than being used for construction of genus-, species- or phylogroup-specific reaction databases, the availability of massive genomics data also allow for integrating comparative genomics analysis (e.g., searching for orthologous genes) into the gap-filling process. Yet, a further investigation of the best use of genomics data in the context of pangenomic gap-filling is beyond the scope of the current work. We have updated the corresponding text in the revised manuscript (see Page 17-18, Lines 324-330):

“Alternative to content removal, the number of false positives can be reduced by limiting the database size for candidate reactions. Though at its infancy, database reduction is a valuable technology that has broad utility for both optimization- and topology-based methods. There are many possible routes of database reduction. First, a universal database can be split into genus-, species-, or even phylogroup-specific databases by aggregating all reactions in GEMs that belong to individual taxa. Despite the pioneering efforts in AGORA models, we are still lacking a large-scale database of high-quality GEMs that cover a wide range of taxonomic diversity.”

Despite the database limitation, we have used the existing databases to create genus- or species-specific databases and tested their utility on gap-filling. First, we constructed a genus-specific database from the BiGG universal reaction database (see Section 4.3 in Supplementary Information for details). For the fermentation product test where CHESHIRE improves over the draft GEMs using the universal database, no improvement was observed for the combination of CHESHIRE and genus-specific databases. Second, the BiGG database contains 58 high-quality *Escherichia coli* GEMs. We explored whether the 58 GEMs help fill the gaps in

a draft model for *E. coli* K-12 MG1655, which was already included in our nutrient utilization and gene essentiality tests. We constructed an *E. coli*-specific reaction database from the 58 GEMs by aggregating reactions. Unfortunately, this species-specific reaction database, when used with CHESHIRE, does not show improvement over draft GEMs (see Fig. R5). One possibility for the failure is that GEMs belonging to the same genus or species might share similar gaps and need to be filled by using reactions from distantly related organisms.

Point 1.7. 7) There are cases of ‘gap-filling resistant’ cases, where none of the current tools can fill a gap. If CHESHIRE can gap fill such cases, it is not only statistically better, but also better ‘in kind’ compared to alternative methods.

Response: We thank Reviewer #1 for this comment. Indeed, substrate utilization and gene essentiality are the examples of “gap-filling resistant” cases for CHESHIRE and the other topology-based methods such as NHP (see our response to Point 1.5 for details). Despite the challenge, CHESHIRE substantially improves the prediction of substrate utilization by *B. subtilis*, while NHP shows no improvement. At least for this example, CHESHIRE is better in kind than alternative topology-based methods. We have updated the corresponding texts in the revised manuscript (see Page 16-17, Lines 306-311):

“Despite the success, we found that substrate utilization and gene essentiality are gap-filling resistant cases for CHESHIRE and, broadly, the topology-based gap-filling methods. Even in this worst scenario, CHESHIRE shows better performance than the competitive method NHP in the prediction of substrate utilization for B. subtilis. Since only 5 GEMs were used in each of the two tests, a comprehensive assessment of CHESHIRE over a larger GEM collection may be more informative of how much CHESHIRE struggles with these tasks.”

Point 1.8. Technical comments:

1) Considering the large number of deposited genomes on public databases and current progress on pangenome analysis there is an emerging trend of generating large-scale metabolic reconstruction which demands scalable and fast gap-filling approaches. Although authors claimed that CHESHIRE is faster and more scalable than the current ML-based approaches, data on computation time and applying the CHESHIRE on a large set of draft networks are not provided in this research.

Response: We thank Reviewer #1 for this valuable comment. We agree that a scalable approach is highly valuable for the large-scale reconstruction of GEMs. In the revised manuscript, we have compared the running time of CHESHIRE with the state-of-the-art topology-based methods, including C3MM and NHP, on the five largest models (based on the number of reactions) from the BiGG database [2]. The five models are Recon3D (*Homo sapiens*), iCHOv1 (*Cricetulus griseus*), iLB1027_lipid (*Phaeodactylum tricornutum* CCAP 1055/1), iCHOv1_DG44 (*Cricetulus griseus*), and RECON1 (*Homo sapiens*) (see Table R1 for their numbers of metabolites and reactions). The running time is obtained for the first type of internal validation in a Mactonish machine with Apple M1 Pro chip and 32 GB memory. Clearly, CHESHIRE is the most computationally efficient method in predicting missing reactions (see Table R2, corresponding to Supplementary Information Table S4). We have updated the corresponding texts in the revised main text (see Page 16, Lines 297-301):

“Compared to previous gap-filling methods, CHESHIRE adopts the concept of hypergraphs with advanced graph convolutional networks to accurately learn the geometrical patterns of metabolic networks and predict missing metabolic reactions without inputs from any experimental data. In addition, CHESHIRE is computationally efficient than C3MM and NHP (Table S4, Supplementary Information Section 3.4).”

and Supplementary Information (see Section 3.4, Lines 285-292):

*“Furthermore, we compared the running time of CHESHIRE with C3MM and NHP on the five largest GEMs (based on the number of reactions) from the BiGG database. We did not consider NVM because of its poor performance in internal validation. The testing GEMs include Recon3D (*Homo sapiens*), iCHOv1 (*Cricetulus griseus*), iLB1027_lipid (*Phaeodactylum tricornutum* CCAP 1055/1), iCHOv1_DG44 (*Cricetulus griseus*), and RECON1 (*Homo sapiens*). The running time is computed based on the first*

set of internal validation in a Mactonish machine with Apple M1 Pro chip and 32 GB memory. As shown in Table S4, among all the three methods, CHESHIRE is the most computationally efficient method in predicting missing reactions.”

Point 1.9. 2) In some cases, gaps are produced by reactions directionality rather than lack of a specific reaction, it seems Hypergraph does not consider reactions directionality. Which might affect the number of added reactions during gap-filling.

Response: We thank Reviewer #1 for pointing this out. Indeed, CHESHIRE does not consider reaction directionality. Other hyperlink prediction methods, except for NHP, treat all reactions as bidirectional too. We have updated the corresponding text in the revised manuscript (see Page 18, Lines 334-340):

“Finally, we should not ignore the possibility that gaps may be filled by altering the directionality of reactions without adding new ones. The current version of CHESHIRE does not consider reaction directionality and thus cannot fill gaps caused by wrong directions. A systematic integration of available thermodynamic data with GEMs will reduce the number of gaps and thus the total number of incorrectly introduced reactions. This deserves dedicated efforts, and we leave it as a future work.”

Point 1.10. 3) Although BiGG universal reactome contains a curated pool of reactions, it does not consider reactions directionality, also it contains same metabolite and reactions with different identifiers. In this regard, did authors curate these inconsistencies before using BiGG universal reactome? If not, how do they justify using an inconsistent pool of reactions for validation of their work?

Response: We thank Reviewer #1 for this valuable comment. In our calculations, we did not curate the metabolite or reaction identifier inconsistencies in the BiGG universal reactome. The BiGG database we used contains 10,393 metabolites (unique IDs) and 16,337 reactions (unique IDs). We found that 255 metabolite IDs (2.45%) have ambiguous names, i.e., names associated with more than one metabolite IDs, and the averaged number of metabolite IDs per unique name is as low as 1.02. This is consistent with what was reported in Pham *et al.* [3] that the averaged number of metabolite IDs per compound name is 1.01. Similarly, 328 reaction IDs (2.01%) have ambiguous reaction formula, i.e., the same reactions associated with more than one identifiers, and the averaged number of reaction IDs per reaction formula is 1.01. Since the duplication of metabolites and reactions only inflate the entire database slightly, the inconsistencies should have minimal impacts on gap-filling. The high quality and low percentage of inconsistencies is exactly the reason why we prefer the BiGG database [2] over the other reactome databases such as ModelSEED [4]. We have updated the corresponding text in Supplementary Information (see Section 4.3, Lines 353-358):

“The resulting BiGG database contains 10,393 metabolites (unique IDs) and 16,337 reactions (unique IDs). We found that 2.45% of metabolite IDs have ambiguous names, i.e., names associated with more than one metabolite IDs. Similarly, 2.01% of reactions have ambiguous reaction formula, i.e., the same reactions associated with more than one identifier. Since the duplication of metabolites and reactions only inflate the entire database slightly, we did not further curate the BiGG universal database to resolve these inconsistencies.”

Point 1.11. 4) For external validation, Authors claimed that by adding the top 200 or 500 reactions with high confidence score will result in a gap-filled model with more precise secreted-metabolites prediction, considering the size of a normal bacterial GEM (800-2000 reactions), it doesn't make sense to add 200/500 reactions during gap-filling, unless there would be more than 500 genes missing from genome annotations, which is unrealistic.

Response: We thank Reviewer #1 for this critical comment. We agree with the reviewer that adding 500 reactions is unrealistic for bacterial genomes which typically have 1,000-2,000 reactions. In our previous submission, we chose 500 because roughly this number of reactions must be included in order to have a significant improvement on the prediction of fermentation products compared to the control approach that randomly adds reactions. The major factor contributing to this large number is the reaction database size, i.e., BiGG universal database has nearly 17,000 reactions and many of them are very similar in terms of their

reactants and products profiles (e.g., may only differ by the co-factor). We found that CHESHIRE encounters difficulty in distinguishing these subtle differences. Eventually, similar reactions will be returned with similar confidence scores. Therefore, any reaction, even a true positive, may be surrounded in the rankings by a number of false positive reactions with similar enzymatic processes. The percentage of false positives thus increases proportionally to the total size of the reaction database. This also explains why the confidence scores of hundreds to thousands of reactions in draft GEMs are so high and fall within a narrow range between 0.999 and 1.0.

In the revised manuscript, we have attempted to reduce the number of reactions added during gap-filling in two different ways. First, we built genus-specific BiGG reaction pools (see Supplementary Information Section 4.3 for details). We found that gap-filling using these pools does not statistically improve the predictions made by intact draft models (see Fig. R4). This might be explained by phylogeny, where closely related microorganisms (e.g., within the same species or genus) may share similar gaps in their genome annotations. Second, we introduced a scoring metric to measure the similarity of candidate reactions to those already in the draft models. The similarity score between two reactions is computed based on the correlation of the corresponding vectors in the stoichiometric matrix. Note that similarity scores of reactions are different from confidence scores that quantify the probability of being present for these reactions. We reason that dissimilar (rather than similar) reactions should be preferentially added since they are complementary to the existing metabolic network and more likely to fill the gaps. Following this strategy, we have successfully reduced the number of reactions that are added to draft GEMs from 500 down to 200. The top 200 reactions are selected by ranking the similarity scores (from lowest to highest) of all reactions whose confidence scores are equal or above 0.9995. With this dual-metric strategy for ranking reactions, CHESHIRE significantly improves the prediction of fermentation products and amino acids compared to intact draft models as well as draft models plus randomly added reactions (see Fig. R9, corresponding to Fig. 3 in the revised manuscript). We have updated the corresponding text in the revised manuscript (see Page 11, Lines 189-197):

“Briefly, CHESHIRE is trained on the entire reaction set of a draft GEM, and candidate reactions (taken from a reaction pool, e.g., the BiGG database [30]) are ranked based on both confidence and similarity scores. The confidence score, returned by CHESHIRE, quantifies the probability of a candidate reaction being present in the GEM. The similarity score measures the maximum correlation between a candidate reaction and all existing reactions in the GEM. Given our rationale that dissimilar reactions are more likely to be functionally complementary to the existing ones, all candidate reactions whose confidence scores ≥ 0.9995 are ranked by their similarity scores (least similar to most similar). The top 200 reactions are added to the draft GEM to produce a gap-filled GEM (Supplementary Information Section 6.1).”

Please also see our responses to Point 1.2 and Point 1.16 on how future studies may address this limitation.

Point 1.12. 5) In regard with external validation, authors only used draft models generated by CarveMe pipeline, though if their methodology is strong enough to fill metabolic gaps efficiently, it should be generalizable to the GEMs generated by other pipelines (ModelSEED, Gapseq, Merlin, etc.), it is highly recommended to diversify testing set not only by randomization of reactions deletion but also by considering GEMs from different sources.

Response: We thank Reviewer #1 for this excellent suggestion. Among all existing pipelines, only CarveMe, ModelSEED, and gapseq generate “ready-to-use” GEMs that allow simulation of organism’s biomass production and metabolic physiology [5]. Gapseq has an innate gap-filling method using a filtered pool of candidate reactions supported by genetic evidences, whose information was not used by CHESHIRE. Therefore, only GEMs generated by CarveMe and ModelSEED can be used for a fair external validation of CHESHIRE.

In the revised manuscript, we have tested the performance of CHESHIRE to fill phenotypic gaps (fermentation products) in both CarveMe- and ModelSEED-reconstructed draft models. In all tests, CHESHIRE outperforms the control approach by randomly adding reactions and a previous topology-based method NHP (see Fig. R9 and Fig. R3, corresponding to Fig.3 in the revised manuscript and Fig. S6 in revised Supplementary Information, respectively). These new results demonstrate that CHESHIRE enables consistent improvement of phenotypic prediction over GEMs reconstructed from different sources. We have updated the corresponding text in the revised manuscript (see Page 12-13, Line 223-229):

“To test whether the improvement of CHESHIRE over CarveMe-reconstructed draft models is generalizable to GEMs from other sources, we performed the same fermentation test on ModelSEED-reconstructed draft models. Similar results were observed, where CHESHIRE-200 improves the predictions over draft models, and draft models plus randomly selected 200 reactions (Fig. S6). NHP, again, shows no improvement. Our results suggest that CHESHIRE enables consistent improvement of phenotypic prediction over GEMs reconstructed from different pipelines.”

Point 1.13. 6) Both internal and external validations are performed on high quality data sets. BiGG models are known as most curated highly consistent models which mostly generated on model organisms with less knowledge gaps. Also, external validation is uniform, in this case, models are generated from high quality reference genomes which subsequently will not be problematic during gap-filling due to a smaller number of missing genes from annotation. Also, all of them have been generated using the same pipeline, which will affect the results in a biased manner.

Response: We thank Reviewer #1 for this critical comment. We agree with the reviewer that even though artificial gaps are introduced in each model, the remaining network connectivity might be more complete compared to those with low-quality models as the starting point. Therefore, in the revised manuscript, we have performed internal validations on 818 AGORA models, which are considered as having intermediate quality between draft models and well-curated models. CHESHIRE still outperforms existing topology-based methods in resolving the artificially introduced gaps in 818 AGORA models (see Fig. R1, corresponding to Fig. S4 in the revised Supplementary Information). The results suggest that the improvement of CHESHIRE over previous methods is consistent and not sensitive to the input GEM quality.

Note that all external validations were performed to gap-fill draft (not curated) models. In the revised manuscript, we have considered both CarveMe and ModelSEED pipelines for draft model reconstruction. On average, the ModelSEED draft models are less predictive of fermentation products and amino acid secretions compared to the CarveMe draft models. Yet, even for the CarveMe draft models, their mean F1 scores are 0.23 and 0.16 for fermentation products and amino acids respectively, suggesting substantial gaps that might result from missing genes. Most importantly, for both phenotypes (fermentation products and amino acid secretions), CHESHIRE improves the predictions over the control approach that randomly adds reactions as well as NHP (the most up-to-date topology-based method before CHESHIRE); see Fig. R9 (corresponding to Fig. 3 in the revised manuscript). All these tests provides compelling evidences that CHESHIRE does not favor a specific reconstruction pipeline.

Point 1.14. 7) Title of the article should be more detailed, the deep learning is a general terminology, it would be more precise if authors consider reforming title based on specific approach they've used (graph convolutional networks).

Response: We thank Reviewer #1 for this excellent suggest. We have changed our title to:

“Teasing out Missing Reactions in Genome-scale Metabolic Networks through Graph Convolutional Networks”

Point 1.15. 8) It is recommended that authors consider a graphical abstract illustration which explains network architecture in detail, it would help readers skim through the provided workflow easier.

Response: We thank Reviewer #1 for this excellent suggestion. We have significantly improved our illustration of CHESHIRE in Fig. R7 (corresponding to Fig. 1 in the revised manuscript), which includes the detailed network architecture of CHESHIRE at both training and prediction phases.

Point 1.16. 9) Judging gapfilling efficiency by capability of fermentation profile is not a strong argument toward defending what authors have claimed in this research, it is known that different enzymes could produce same fermentation metabolism from different substrates. Therefore adding reaction from a universal reaction pool (when reaction pool is not strain/species or even genus specific reaction pool) might result in right phenotype but with wrong reactions. For example, false-positive phenotype of *Anaerobutyricum hallii* draft network (15) might be simply solved by checking the directionality of reaction instead of adding 500 reactions to make it

predict better. Or in worst case scenario adding an isozyme of lactate dehydrogenase with different co-factor might also be helpful for solving such a simple problem.

Response: We thank Reviewer #1 for this very insightful comment. Indeed, with a universal reaction pool, CHESHIRE risks of adding incorrect reactions, e.g., alternative pathways or reactions with different enzymes, co-factors and substrates, that happen to result in the right phenotype. We have acknowledged this limitation in the Discussion section of the revised manuscript (see also our response to Point 1.2). We have further tested the approach of using the genus-specific reaction pool (see Supplementary Information Section 4.3), as suggested by the reviewer. However, the use of genus-specific reaction databases does not improve the predictions of fermentation products by CHESHIRE compared to draft models (see Fig. R4).

The reviewer mentions an alternative gap-filling strategy by changing reaction directionality. We fully agree that it may result in the right phenotype by only changing reaction directionality without adding new reactions. However, the selection of which reactions for the change of directionality might need to be guided by knowing the phenotype as a priori knowledge. This is different from CHESHIRE which infers missing reactions without knowing any experimental data. It is possible that the lactate phenotype of *Anaerobutyricum hallii* draft network may be solved by changing reaction directionality. Yet, developing an algorithm to pick the right reactions and assigning their directions, especially without any guidance from data, is a highly nontrivial task and deserves a dedicated project on its own. This comment on reaction directionality overlaps with Point 1.9: see our response to this comment for more information.

We also agree with the reviewer that gap-filling efficiency should be evaluated based on research objectives. CHESHIRE not only facilitates the model curation step towards building a high-quality GEM, but also it fulfills a pressing need for rapid prediction of phenotypes given bacterial genomic sequences before any experiment is done. Rapid phenotypic prediction plays an important role of *in silico* screening, i.e., phenotypes can be simulated from draft GEMs. Here we have demonstrated that draft models with CHESHIRE-predicted missing reactions added back enables more accurate predictions. In this context, judging gap-filling efficiency by capability of predicting the right phenotypes is appropriate, and whether CHESHIRE predicted the right reactions becomes less concerned (and it has been systematically tested during internal validations). We have updated the corresponding text in the revised manuscript (see Page 4, Lines 48-56):

“However, experimental data is not readily available for non-model organisms, thus limiting the utility of those tools. For example, most intestinal organisms are considered “uncultivable” and their functions remain unknown [19]. Even for cultivable organisms, high-throughput phenotypic screening, i.e., searching for organisms with desired phenotypes, relies on the analysis of microbial extracts or genetic modifications, which can become complicated, time-consuming, and expensive. Given the increasing availability of cultivable organisms and their genomes, there is a pressing need for rapid and accurate in silico predictions of metabolic phenotypes solely from genomic sequences. Even though the predictions are theoretical, downstream experimental validations could be much less resource-demanding.”

Finally, we thank Reviewer #1 again for reviewing our manuscript and her/his very insightful and constructive comments, which have helped us significantly improve the quality of our manuscript. We hope our responses above have addressed all her/his concerns in a satisfactory manner.

Response to Reviewer #2

Point 2.0. The manuscript presents a deep learning-based method — CHEbyshev Spectral Hyperlnk pREdictor (CHESHIRE) — to predict missing reactions of genome-scale metabolic models (GEMs) purely from the metabolic network topology. The authors claim that the method outperforms other topology-based methods in 108 high-quality GEMs. Actually, the authors correctly state in the abstract that they offer “compelling evidence that CHESHIRE outperforms other topology-based methods,” as this kind of method can only be compared by simulation experiments, and the results of the comparison still depend strongly on the tested conditions.

The manuscript does a perfect job of describing the problem, the current procedures, and the challenges of the gap-filling process.

Response: We thank Reviewer #2 for reviewing our manuscript and the overall positive assessment of our work. Next, we address each of her/his comments in order.

Point 2.1. The gap-filling process involves adding reactions in a metabolic model to have the model consistent with observed phenotypes, such as growth assays (on various substrates), metabolite production, and true-positive and true-negative predictions in knock-out experiments. In general, there are two main approaches to introducing reactions that improve model consistency: (1) optimization-based approaches that seek to maximize model consistency with observed phenotypes (production or consumption of specific metabolites, growth, etc.) by introducing, from a given database, a min number of reactions (although one can identify multiple alternative minimal or great sets of reactions), and (2) network topology methods of which, the latest and improved methods use ML methodologies. Both of these approaches use precisely the same starting inputs: an initial draft model and a reference database of reactions used for gap-fill.

The network topology methods first introduce many reactions, depending on the technique and user-defined parameters. They then use a set of phenotypic data to test if the gap-filled model can simulate these phenotypes. However, there are still many reactions that are now part of the model for which there is no genetic or physiological justification for their presence. In the optimization-based approach, the phenotypic data are used in advance. The methods guarantee that the reactions added to the model are necessary (even as an alternative) for simulating the observed phenotype. In this process, no reaction is added, which does not contribute to the observed phenotype. In general, both methods could provide the result, the computational effort could be comparable, and they both involve a manual curation process, as the last process in the topology-based approach is a manual curation. If the authors agree with the above statements, then we would agree that some of their statements about the lack of manual curation and the no need for phenotypic data in ML network-topology-based methods are strong, and they can be misleading.

Response: We thank Reviewer 2 for this critical comment. Indeed, both network topology-based and optimization-based methods use a draft model and a reaction database for gap-filling. However, topology-based methods, including CHESHIRE, are fully **unsupervised**, i.e., automatically introducing many reactions without the need of knowing any genetic, physiological, or phenotypic data. By contrast, optimization-based methods are **supervised** and requires experimental data to start with. Although the two approaches address the same problem, it is not fair to compare a supervised method and an unsupervised method. In fact, when phenotypic data is available, we recommend using optimization-based methods which, as the reviewer mentions, only add a minimal number of reactions necessary for simulating the phenotype. The biggest advantage of our method, compared to optimization-based methods, is the rapid prediction of phenotypes as well as reactions potentially missing from the draft model in the absence of any experimental data (see our response to Point 1.16 from Reviewer 1). This advantage does comes with a cost, i.e., it may introduce many unjustified reactions, which eventually needs manual curation to trim the false-positives if the ultimate goal is to build a high-quality GEM. Since the manual curation step is not part of our pipeline (or any other topology-based methods), our statements of the lack of manual curation and the no-need for phenotypic data remains technically valid. We admit that this statement might be confusing without a detailed explanation and comparison of the topology-based approach and the optimization-based approach. To avoid misleading readers, we have updated the corresponding text in the revised manuscript (see Page 15-16, Line 279-291):

“Optimization-based GEM gap-filling has been long considered as a process of fitting a GEM to observed data [44]. This problem is typically formulated by a mixed-integer linear programming that minimizes the number of added reactions under the constraint that the observed phenotypes are satisfied. Therefore, the majority of GEM gap-filling methods falls short of predicting metabolic gaps in

both network connections and functions without knowing experimental phenotypes a priori. FastGapFill, as one of a few exceptions, fits a specific task of gap-filling to resolve dead-ends and blocked reactions [21]. Previous studies [24, 45, 46] have shown that FastGapFill exhibits a poor performance in filling artificially introduced gaps. Although gap-filling with experimental data is critically important, it is limited to understanding the gene-reaction-phenotype mappings in conditions where the data was collected. The environmental conditions are combinatorially complex; as a theoretical tool, the primary purpose of GEMs is to rapidly offer theoretical predictions of metabolic activities over a large array of environmental conditions where data has not been collected.”

Point 2.2. Another major issue appears to exist in the comparison between the optimization-based methods and the CHESHIRE method (and the topology-based methods), the conclusions, and the statements about the results of the comparison.

It is unclear if the authors performed the CarveMe reconstruction themselves or if they used the models in the CarveMe database. Even if the authors used the CarveMe workflow and reconstructed the so-called draft models, did they use only the growth as a phenotype for gap-fill within the CarveMe workflow, or did they use the fermentation data to obtain the draft? I assume that the authors used only the growth as a phenotype for gap-fill, which is the default procedure for the published methodology of CarveMe. In this case, I would argue that the CarveMe draft models did not “know the experimental phenotypes a priori.” In this respect, their performance is very good since no reaction is added except for a minimal set for growth. To perform a fair comparison, the authors should compare the models from their method with models derived using the optimization gap-fill method in CarveMe, which uses the fermentation data as input in advance, since the author claim that their method does better than the optimization-based methods that use the phenotypic data as a priori knowledge. It is also unclear if the authors used the same (extracellular) media as in the CarveMe protocol. The definition of media is crucial for model reconstruction and validation, and the topology methods do not consider it. The authors here do not clearly define it and justify the one they use.

Response: We thank Reviewer #2 for these great questions. We apologize for not explaining the comparison of topology- and optimization-based approaches adequately; the reviewer had an impression that we claimed that our method is superior to optimization-based methods. However, we were not intended to claim this. As we have explained in our response to Point 2.1, topology-based approaches are unsupervised (without phenotypic data as input) and optimization-based approaches are supervised (require phenotypic data as input). Therefore, their performances are not comparable. Instead, our manuscript focused on the comparison between CHESHIRE and other topology-based approaches such as NHP and C3MM.

We also apologize for not being very clear about the details of our modeling approach. The reviewer is correct. We generated the CarveMe models using the standard pipeline and only used growth phenotype for gap-filling. The reviewer is also correct that the CarveMe models does not know the experimental phenotype as a prior. When we compared the CHESHIRE-predicted models to the draft models, both models include the minimum set of reactions added by CarveMe for gap-filling the growth phenotype. Even though CarveMe-added reactions might by accident help fill other phenotypic gaps in the validation set, these reactions were equally added to all models and the comparison of their performances remains fair. We have added these details in Supplementary Information (see Section 6.1, Line 447-449):

“All draft GEMs were reconstructed using the standard CarveMe [31] or ModelSEED [34] pipelines. Only growth phenotypes were used for the built-in gap-filling algorithm in each pipeline.”

In general, the environment in the simulations was designed to mimic the culture media used in the experiments. We have performed four different phenotypic tests in the revised manuscript: two tests (amino acid secretions and substrate utilization) used M9 minimal medium whose composition can be found in the CarveMe media database. The culture media composition used in the other two tests were obtained from other sources. In the revised SI, we have added a detailed description of different culture media used in the external validation (see Supplementary Information Section 6.2, Lines 469-489):

“6.2. Culture Media Compositions. The culture media compositions used for growth simulations were determined to reproduce the experimental conditions under which phenotypes were measured. While the dataset of fermentation product test result from multiple experiments whose culture media can vary, we followed the same strategy as described in Zimmermann et al. [28] and assumed that all experiments

were performed under the same growth medium. We further adopted the fermentation test medium composition and their maximally allowed fluxes developed in the same study (accessible from <https://github.com/Waschina/gapseqEval>). For the amino acid secretion test, M9 minimal medium (with glucose) was used. Glucose has a maximum uptake rate of 10 mmol/gDW/h and all other compounds in the medium were unconstrained. For the substrate utilization test, GEMs were also constrained to the same M9 minimal medium, where the default sources of carbon, nitrogen, sulfur and phosphorus are glucose, ammonia, sulfate, and phosphate, respectively. For *Shewanella oneidensis*, the default carbon source is DL-lactate. To simulate growth on each substrate in Biolog arrays, the default source with the same type of the substrate (i.e., carbon, nitrogen, sulfur, and phosphorus) in the M9 minimal medium was replaced with the substrate. The maximum uptake rate for all Biolog substrates is 10 mmol/gDW/h and all other compounds in the M9 medium are unconstrained. We downloaded the M9 recipe from the github repository of CarveMe (accessible from <https://github.com/cdanielmachado/carveme>). For gene essentiality test, the culture media compositions were available from the same github repository: M9 minimal medium (with glucose) for *E. coli*, M9 minimal medium (with succinate) for *P. aeruginosa*, LB medium for *B. subtilis* and *S. oneidensis*, and complete medium (all compounds with exchange reactions are allowed to be uptaken) for *M. genitalium*. All compounds in the culture media were unconstrained."

Point 2.3. Finally, I think that some major issues reduce the significance and broader impact of the manuscript.

First, the improvement of the method relative to other ML, network-topology-based methods is insignificant and only provided as evidence. A larger set of models and phenotypes should be used to test that at least improvement is consistent and quantitatively better.

Response: We thank Reviewer #2 for this critical comment. In the revised manuscript, we have shown clear evidence that CHESHIRE outperforms other existing topology-based methods in both internal validation and external validation over different collections of GEMs, validation strategies, and metabolic phenotypes (see Fig. R8 and Fig. R9, corresponding to Fig. 2 and Fig. 3 respectively in the revised manuscript). The statistical significance has been assessed by the two-sided paired-sample t-test for commonly used performance evaluation metrics, including AUROC/AUPRC, precision, recall and F1 scores.

We agree with the reviewer that a broader test is needed to assess the consistency of improvement. In the revised manuscript, we have expanded both internal and external validations. For internal validation, we have added comparisons of CHESHIRE and the existing topology-based methods (NHP and C3MM) to fill artificially introduced gaps on 818 AGORA models of gut bacteria [1] (see Fig. R1, corresponding to Fig. S4 in Supplementary Information). We have updated the corresponding text in the revised manuscript (see Page 9, Lines 159-161):

"We also performed the same type of internal validation on a larger set of metabolic networks, i.e., 818 AGORA models from the Virtual Metabolic Human database [8], and observed similar results (Fig. S4)."

For external validation, we have showed that the improvement of CHESHIRE on filling the gaps of fermentation products can be generalized to amino acid secretions (see Fig. R9, corresponding to Fig. 3 in the revised manuscript). We have added the results of this new test in the revised manuscript (see Page 13, Lines 230-240):

"The fermentation test comprises nine metabolites that are very close to central carbon pathways. To test if CHESHIRE can fill other types of gaps, we assessed CHESHIRE for predicting secretions of amino acids, substrate utilization for growth, and gene essentiality. The dataset of amino acid secretions measures production profiles of 20 amino acids for 25 bacterial GEMs (Table S2, Supplementary Information Section 4.5). This dataset is highly unbalanced with 478 positive phenotypes and 22 negative phenotypes. Similar to the fermentation test described above, CHESHIRE-200 outperforms Random-200 (F1 score: $P < 10^{-5}$, two-sided paired-sample t-test), and NHP-200 shows no improvement at all (Fig. 3f-i, Fig. S5). In particular, CHESHIRE increases the correct predictions of 67 amino acid secretions that are knowledge gaps in draft GEMs. Despite the significant improvement, the recall remains very low at about 20%, suggesting many remaining false-negative gaps."

Since the reviewer also mentions growth and gene-knockout phenotypes in Point 2.6, we have tested the performance of CHESHIRE on 5 GEMs whose substrate utilization and gene essentiality data are publicly

available. It turns out that CHESHIRE, as well as the existing topology-based method NHP, struggle with filling the gaps and no improvement is observed beyond the draft models (see Fig. R5, corresponding to Fig. S7 in the Supplementary Information). It is important to note, CHESHIRE is the first method, among all existing topology-based methods, that has been rigorously tested over different collection of GEMs and metabolic phenotypes. Existing topology-based methods such as NHP were only tested using an internal validation strategy on artificially introduced gaps but not on real biological data. We have updated the corresponding texts in the revised manuscript:

*“Each of the substrate utilization and gene essentiality tests contains 5 GEMs (Table S3, Supplementary Information Section 4.6 and 4.7). The utilization of various carbon-, nitrogen-, phosphorus-, and sulfur-substrates for growth were tested using Biolog phenotype arrays [38] in high-throughput manner. Essential genes were identified using gene knockout experiments and a gene is essential if its deletion is lethal and causes cell death. When tested on both datasets, CHESHIRE, however, fails to fill gaps in nearly all 5 GEMs, except for the growth phenotypes of *Bucillus subtilis* where CHESHIRE-200 increases the F1 score of CarveMe draft model from 0.58 to 0.87 (Fig. S7). Notably, NHP fails in all the tests, including the growth phenotypes of *B. subtilis*.” (see Page 13-14, Lines 241-248)*

*“Most importantly, CHESHIRE has been validated on realistic biological datasets. To our best knowledge, such benchmark has not been performed for previous topology-based gap-filling methods. We showed that CHESHIRE significantly improves the phenotypic predictions of fermentation products and amino acids secretions over a total of 49 draft GEMs reconstructed from a mostly used automatic reconstruction pipeline CarveMe [11]. Despite the success, we found that substrate utilization and gene essentiality are gap-filling resistant cases for CHESHIRE and, broadly, the topology-based gap-filling methods. Even in this worst scenario, CHESHIRE shows better performance than the competitive method NHP in the prediction of substrate utilization for *B. subtilis*. Since only 5 GEMs were used in each of the two tests, a comprehensive assessment of CHESHIRE over a larger GEM collection may be more informative of how much CHESHIRE struggles with these tasks.” (see Page 16-17, Lines 301-311)*

Point 2.4. Second, the models derived from the CHESHIRE method have an order of 500 reactions which might have no physiological relevance. I am afraid that the community and the users of these models would like to have some justification for the excessive (useless) number of reactions. How could the authors trim these reactions without going through a manual curation? Unless they suggest a manual curation, which goes against the main claimed advantage of their method (no need for manual curation). Although, as we discussed above, the network-topology methods still involve much effort in manual curation.

Response: We thank Reviewer #2 for this valuable comment. The reviewer is correct that adding 500 reactions may be too much and risk of incorporating false positives. This point is essentially the same as Point 1.11 raised by Reviewer 1. Please see our response to Point 1.11 for details. Briefly, we have successfully reduced the number of added reactions from 500 to 200 by using a combination of confidence scores (indicating the confidence of a candidate reaction being present) and similarity scores (indicating the averaged similarity between a candidate reaction and those in the draft models) to rank the candidate reactions. The similarity score helps us automatically trim reactions, without going through a manual curation. We have updated our approach for ranking reactions in the revised manuscript (see Page 11, Lines 189-197):

“Briefly, CHESHIRE is trained on the entire reaction set of a draft GEM, and candidate reactions (taken from a reaction pool, e.g., the BiGG database [30]) are ranked based on both confidence and similarity scores. The confidence score, returned by CHESHIRE, quantifies the probability of a candidate reaction being present in the GEM. The similarity score measures the maximum correlation between a candidate reaction and all existing reactions in the GEM. Given our rationale that dissimilar reactions are more likely to be functionally complementary to the existing ones, all candidate reactions whose confidence scores ≥ 0.9995 are ranked by their similarity scores (least similar to most similar). The top 200 reactions are added to the draft GEM to produce a gap-filled GEM (Supplementary Information Section 6.1).”

We agree with the reviewer that manual curation is needed as a downstream step of CHESHIRE to further trim false positives. The similarity score we introduced here is an automatic curation process based on a reasonable assumption. While we successfully reduced the number of reactions derived from CHESHIRE to 200, it is still a heavily manual task to further trim the remaining reactions. This task usually need domain

knowledge, which is beyond the scope of topology-based gap filling. We have updated the corresponding text in the revised manuscript to emphasize this point (see Page 17, Lines 312-323):

“Although CHESHIRE advances phenotypic predictions, the use of a universal pool and the top 200 reactions risk of adding reactions that do not exist (false positives). Correct predictions of phenotypes do not necessarily mean correct inference of missing reactions. It is likely that different enzymes carry the same metabolic functions (e.g., fermentation metabolism) from different substrates. While adding reactions is expedited by increasingly advanced machine learning methods, it is still a heavily manual task to trim reactions that are wrongly added. This challenge, termed as content removal, has been recognized as one of the two fundamental bottlenecks for GEM quality improvement [47]. This bottleneck is largely overlooked and highly time-consuming: a significant amount of time is required to identify which reactions should be removed. Initial community-driven efforts have been made to mitigate this challenge, including building new GEM quality standards such as MEMOTE [48] and developing novel frameworks for GEM quality assessment and improvement.”

Point 2.5. Third, the phenotypic data comprises nine metabolites that are very close to central carbon pathways. The methods should be tested for various produced metabolites (or phenotypes that involve metabolites) distributed across different classes of metabolites (amino acids, lipids, secondary metabolites, terpenoids, etc.). These compounds are in complex pathways, and the gap-filling of these pathways is a very challenging problem for any gap-filling method.

Response: We thank Reviewer #2 for this valuable comment. We agree with the reviewer that gap-filling compounds in complex pathways are very challenging. To challenge CHESHIRE, we contacted an European team to request a collection of 25 bacterial genomes with amino acid secretion profiles they recently published on *Current Biology* [6]. On this new dataset, CHESHIRE greatly improves the predictions of amino acid secretions against the control approach (by adding random reactions) as well as the existing topology-based method NHP (see Fig. R9f-i, corresponding to Fig. 3f-i in the revised manuscript). We are not aware of similar large-scale public datasets of paired genomes and phenotypes for other complex compounds mentioned by the reviewer. Gap-filling for these compounds will be challenging though and even impossible if any reaction of their biosynthetic pathways is missing from the universal BiGG database [2]. One such example is valeric acid (5-carbon short-chain fatty acid) whose biosynthetic pathway (odd-chain elongation) is incomplete in the current BiGG version. We have updated the corresponding text in the revised manuscript (see Page 13, Lines 230-240):

“The fermentation test comprises nine metabolites that are very close to central carbon pathways. To test if CHESHIRE can fill other types of gaps, we assessed CHESHIRE for predicting secretions of amino acids, substrate utilization for growth, and gene essentiality. The dataset of amino acid secretions measures production profiles of 20 amino acids for 25 bacterial GEMs (Table S2, Supplementary Information Section 4.5). This dataset is highly unbalanced with 478 positive phenotypes and 22 negative phenotypes. Similar to the fermentation test described above, CHESHIRE-200 outperforms Random-200 (F1 score: $P < 10^{-5}$, two-sided paired-sample t-test), and NHP-200 shows no improvement at all (Fig. 3f-i, Fig. S5). In particular, CHESHIRE increases the correct predictions of 67 amino acid secretions that are knowledge gaps in draft GEMs. Despite the significant improvement, the recall remains very low at about 20%, suggesting many remaining false-negative gaps.”

Point 2.6. Finally, the models derived from the CHESHIRE, and similar methods, should be tested for their performance against growth assays and knock-out experiments, using the metrics used in these studies and model testing and validation (such as the overall accuracy and the Matthews Correlation Coefficient). These essential metrics will also evaluate the number of (mainly) false-positive predictions due to the large number of reactions introduced in the network.

Response: We thank Reviewer #2 for this critical comment. This comment overlaps with Point 1.5 from the Review 1: see our response to Point 1.5 for the additional performance tests and results. Briefly, we have tested CHESHIRE and NHP for their performances against nutrient utilization (growth assay) and gene essentiality (knock-out experiments) data. Although both methods struggle with the tests (see Fig. R5, corresponding to Fig. S7 in the revised Supplementary Information), CHESHIRE, not NHP, greatly improves the F1 score over the draft model in the Biolog test of a single bacterium *Bacillus subtilis*.

Additionally, we have showed overall accuracy and Matthew's correlation coefficient, as recommended by the reviewer, for predictions of fermentation products and amino acid secretions. As shown in Fig. R2 (corresponding to Fig. S5 in the revised Supplementary Information), CHESHIRE still outperforms NHP in terms of those two metrics.

Point 2.7. The final suggestion in the discussion, about the use of CHESHIRE and similar methods for database reduction also used from the optimization methods, is very valuable and could have a broader impact. But it will require a specific and well-designed investigation.

Response: We thank Reviewer #2 for this excellent suggestion. We fully agree with the reviewer that reducing the size of the reaction database is of critical importance to all gap-filling methods, including CHESHIRE. As the reviewer clearly pointed out, this requires a specific and well-designed investigation. During the revision process, we have made initial efforts towards this goal by splitting the BiGG universal database into genus-specific reaction pools (see Supplementary Information Section 4.3 for details). However, this strategy fails to gap-fill the fermentation products (see Fig. R4). Since each genus-specific pool was built by aggregating GEMs that belong to the same genus, it is possible that all models within a genus may share the same missing reactions whose gap-filling requires introduction of new reactions from distantly related genera. Given the broad impacts of database reduction, we have discussed several reduction strategies in the revised manuscript (see Page 17-18, Lines 324-341):

“Alternative to content removal, the number of false positives can be reduced by limiting the database size for candidate reactions. Though at its infancy, database reduction is a valuable technology that has broad utility for both optimization- and topology-based methods. There are many possible routes of database reduction. First, a universal database can be split into genus-, species-, or even phylogroup-specific databases by aggregating all reactions in GEMs that belong to individual taxa. Despite the pioneering efforts in AGORA models, we are still lacking a large-scale database of high-quality GEMs that cover a wide range of taxonomic diversity. Second, GapSeq [12] points out a promising direction to use genomic information for reducing a universal database to a small subset of reactions supported by gene annotations. By lowering the threshold for sequence homology, comparing protein domains and sequence signatures, and mapping content to distantly related organisms, more genes and candidate reactions can be annotated [47]. Finally, we should not ignore the possibility that gaps may be filled by altering the directionality of reactions without adding new ones. The current version of CHESHIRE does not consider reaction directionality and thus cannot fill gaps caused by wrong directions. A systematic integration of available thermodynamic data with GEMs will reduce the number of gaps and thus the total number of incorrectly introduced reactions. This deserves dedicated efforts, and we leave it as a future work.”

Finally, we thank Reviewer #2 again for reviewing our manuscript and her/his very insightful and constructive comments, which have helped us significantly improve the quality of our manuscript. We hope our responses above have addressed all her/his concerns in a satisfactory manner.

Response to Reviewer #3

Point 3.0 Chen, Liao and Liu present a method to predict missing reactions in genome-scale metabolic networks (GEMs) through deep learning based on CHEbyshev Spectral Hyperlink pREdictor (CHESHIRE). This method is based on network topology: given a metabolic network represented as metabolite-reaction incidence matrix, it learns the network structure, and then for a given reaction provides a score indicating how likely this reaction is part of the input network. The authors demonstrate the better performance of their method compared to the other deep learning-based and topology-based gap filling methods on an “internal” validation dataset, where the removed reactions are known, and an “external” validation dataset, where the missing reactions are not known. For the latter, the performance metric is the predicted fermentation phenotype of 9 metabolites, which is compared to experimentally determined phenotypes. The authors also provide biological interpretations of the gap-filled reactions that help to recapitulate the experimentally determined phenotype.

In general, the method is promising, and the application of deep learning in genome-scale metabolic modelling will be interesting for a broader audience interested in machine learning applications in biology, mathematical modelers, and system biologists. However, there is a number of issues that need to be clarified or addressed in order to assess the utility of this method for gap filling of metabolic networks in biological applications.

Response: We thank Reviewer #3 for reviewing our manuscript and her/his positive assessment on the broad application of our method. Next, we address each of her/his comments in order.

Point 3.1. Major comments:

1) From Figure 1d,e and model architecture and training description on pages 7 and 8 it does not become fully clear how the model is trained and what exactly the input data is. On page 5, it is written that CHESHIRE takes a metabolic network and a pool of candidate reactions as input, and produces confidence scores for the candidate reactions as output. However, the input on Figure 1d only depicts metabolic network, and misses the candidate reactions.

Response: We thank Reviewer #3 for this critical comment. We apologize for not describing CHESHIRE clearly in the previous version of our manuscript. In the revised manuscript, we have improved Fig. 1, which now highlights both the training and prediction phases of CHESHIRE (attached as Fig. R7). In the training phase, CHESHIRE takes the incidence matrix of the metabolic network and a decomposed graph built from the metabolic network (with negative reactions) as input. After learning the model parameters (from backpropagation), CHESHIRE takes the incidence matrix of the original metabolic network and a decomposed graph built from candidate reactions from a reaction pool and outputs confidence scores for candidate reactions. We have updated the corresponding texts in the revised manuscript:

“CHESHIRE only requires a metabolic network for training and outputs confidence scores for candidate reactions from a reaction pool.” (see Page 5, Lines 79-81)

“CHESHIRE takes the incidence matrix of the hypergraph and a decomposed graph (built from the hypergraph of existing or candidate reactions) as input. The former contains boolean values indicating the presence or absence of each metabolite in each reaction. The latter consists of fully connected subgraphs (each subgraph represents a reaction with all its metabolites connected) formed by positive and negative reactions during training and by candidate reactions during prediction (Fig. 1c and d). Positive reactions are those existing in the metabolic network, while negative reactions are fake (do not exist) and created for model-balancing purposes (often referred to as negative sampling). Note that only positive reactions are used to construct the incidence matrix.” (see Page 6, Lines 93-101)

Point 3.2. On page 7, it is written that the pooling step computes reaction features, does this mean for reactions from the incidence matrix? How will the reaction features be computed for the pool of candidate reactions? Is incidence matrix also created for them?

Response: We thank Reviewer #3 for this critical comment. The incidence matrix of the metabolic network is only used in feature initialization to generate initial metabolite features. The pooling step computes reaction features for reactions represented by fully connected subgraphs in the decomposed graph (during both training and prediction). Candidate reaction features are computed in the same manner as the training reactions, i.e., for each candidate reaction (represented by a fully connected subgraph in the decomposed graph), aggregating the feature vectors of its metabolites (represented by nodes in the subgraph). Therefore, the

incidence matrix of candidate reactions is not needed. We have updated the corresponding text in the revised manuscript (see Page 7, Lines 109-112):

“For pooling (i.e., integrating node- or metabolite-level features into hyperlink- or reaction-level representation), we utilize graph coarsening methods to compute a feature vector for each reaction (represented by a fully connected subgraph in the decomposed graph) from the feature vectors of its metabolites.”

Point 3.3. 2) Importantly, what is the loss function that is optimized during the training step?

Response: We thank Reviewer #3 for this critical comment. We apologize for not describing the loss function clearly. CHESHIRE uses the same loss function as proposed in [7], which is defined as

$$\text{Loss} = \frac{1}{|\mathcal{E}|} \sum_{e \in \mathcal{E}} \sigma \left(\left(\frac{1}{|\mathcal{F}|} \sum_{f \in \mathcal{F}} S_f \right) - S_e \right),$$

where \mathcal{E} is the set of positive hyperlinks, \mathcal{F} is the set of negative hyperlinks, and $\sigma(\cdot) = \log(1 + \exp(\cdot))$ is the logistic function. The loss function maximizes the hyperlink scores from the positive hyperlink set \mathcal{E} to be higher than the average score of the hyperlink scores in the negative hyperlink set \mathcal{F} [8]. The definition of the loss function can be found in Supplementary Information Section 3.3. We didn't include it in the main text because we want to improve the readability of the main text to typical life scientists.

Point 3.4. On page 8, it is written that the input model is enhanced with negative reactions (“fake” reactions added in the same amount as real reactions). Are these negative reactions also included as input, and is the model provided with the information on which reactions are true, and which are false? It seems like this is a crucial step in model training, but it is not described neither in Figure 1e nor explicitly in the text.

Response: We thank Reviewer #3 for this critical comment. Negative reactions are also included as the input of CHESHIRE from the decomposed graph during the training phase, as mentioned in our response to Point 3.1. CHESHIRE is also provided with the information on which reactions are true, and which are false for updating its parameters. This information has been reflected in the Fig. R7e (corresponding to Fig. 1e in the revised manuscript). To make this point clear, we have updated the corresponding text in the revised manuscript (see Page 7, Lines 116-119):

“In the training phase, the resulting scores are compared to the target scores (one for positive reactions and zero for negative reactions) with a loss function for updating the model parameters (Fig. 1e, Supplementary Information Section 3.3).”

Point 3.5. 3) On page 9, it is written that positive and negative reactions from each model are split into 60% training (metabolic network to be gap-filled) and 40% testing (unseen candidate reactions). Has these procedure been repeated multiple times for different splits in 60% and 40%?

Response: We thank Reviewer #3 for raising this question. Yes, we randomly split the positive and negative reactions into 60% training and 40% testing for 10 times. We have updated the corresponding text in the revised manuscript (see Page 8, Lines 133-134):

“For both types, metabolic reactions in a given GEM were first split into a training set and a testing set over 10 Monte Carlo runs.”

Point 3.6. And what about validation, how were the model parameters tuned during the training phase? Was subset of the 60% used for optimization of model parameters?

Response: We thank Reviewer #3 for this critical comment. We used a universal hyperparameter set for each hyperlink prediction method in internal validation. And, all 60% of the reactions were used for training without creating a subset for tuning/optimizing hyperparameters.

The universal hyperparameter sets are chosen empirically before testing. For example, in CHESHIRE, the two hidden dimensions are set to 256 and 128, the learning rate is set to 0.01, the Chebyshev filter size is set to 3, and the dropout probability is set to 0.1. Universal hyperparameters can provide a conservative estimate of the algorithm performance. As a matter of fact, the performance of CHESHIRE (and NHP, NVM) with a pre-selected universal hyperparameter set is close to that using grid search in our case. We have performed a comparison test on several randomly selected BiGG models. For the universal hyperparameter setting, we have randomly split the total reactions into 60% training and 40% testing for 10 times. For the grid search setting, we have randomly split the total reactions into 60% training, 20% validation, and 20% testing for 10 times. The universal hyperparameters and grid search ranges for CHESHIRE can be found in Table R3. The test results (with metric AUROC) can be found in Table R4, where the performances of the two settings are in fact very close. We also observed similar results for NHP and NVM since both share a similar architecture with CHESHIRE. Lastly, C3MM has only one hyperparameter, which is the latent space dimension. It was fixed to 30 in the original paper [9]. Thus, we also used 30 in our simulation. We have updated the corresponding texts in the revised Supplementary Information:

“The key hyperparameters of CHESHIRE are the encoder feature dimension, the graph convolutional feature dimension, the Chebyshev filter size, the dropout probability, and the learning rate. We used a universal hyperparameter set for CHESHIRE during internal and external validations. We found that the performance of CHESHIRE with a pre-selected universal hyperparameter set is close to that obtained by grid search. This implies that CHESHIRE is not sensitive to these hyperparameters. Therefore, we decided to use a universal hyperparameter set for all the GEMs (which can also save a great amount of computational resources). The encoder feature dimension, the graph convolutional feature dimension, the Chebyshev filter size, the dropout probability, and the learning rate are set to 256, 128, 3, 0.1, and 0.01, respectively.” (see Supplementary Information Section 3.5, Lines 293-301)

“We intended to fairly compare CHESHIRE with other approaches including NHP, C3MM, and NVM during internal validation. Similar as CHESHIRE, NHP and NVM are also not sensitive to their hyperparameters. We set the Node2Vec feature dimension to 256, which is consistent with the encoder dimension in CHESHIRE. The walk length and the number of walks per node were set to 80 and 10 in Node2Vec (default values in the Node2Vec Python package [32]), respectively. Additionally, we set the feature dimension of the graph neural network in NHP to 128, which is also consistent with the dimension of CSGCN in CHESHIRE. The learning rate of NHP and NVM is set to 0.01. For C3MM, we used the same latent space dimension 30 as used in the C3MM paper [10].” (see Supplementary Information Section 5.1, Lines 389-397)

Point 3.7. 4) On page 8, it is written that negative reactions are not mandatory for algorithm testing. But in this section, reactions from the model (40%) are provided for testing, so if only positive reactions will be in the candidate pool, what is the point of assessing performance without negative reactions?

Response: We thank Reviewer #3 for pointing this out and we apologize for this confusion. We did include negative reactions in the candidate reaction pool for evaluating the performance of predicting missing reactions during the first type of internal validation. We have included a schematic plot of this validation process (see Fig. R8a, corresponding to Fig. 2a in the revised manuscript) and updated the corresponding text in the revised manuscript (see Page 8, Lines 138-141):

“In the first type of internal validation, the training and testing sets of positive reactions and their derived negative reactions were combined and used for training and testing, respectively. For fair comparison, we also introduced negative reactions to the testing set of C3MM.”

Point 3.8. 5) For internal validation, providing 40% of model reactions and artificially created negative counterparts as candidate reaction pool seems to be over-optimistic, since in real application the pool of candidate reactions will be much bigger and much less balanced in terms of positive and negative classes (as is the case with external validation). Since all the models used for internal training are curated BiGG models, it would be interesting to see how CHESHIRE and other methods perform if the candidate reaction pool is not coming from the same model with some negative reactions added, but agnostically use BiGG database pool as for external validation, and demonstrate whether CHESHIRE is able to identify original reactions.

Response: We thank Reviewer #3 for this critical comment. We totally agree with the reviewer that using the candidate reaction pool consisting of artificially created negative counterparts is over-optimistic. In addition to the first type of internal validation (as mentioned), we have followed the reviewer's suggestion by utilizing real candidate reactions to perform internal validation. We consider two types of reaction pools here, which are genus-specific reaction pools and the entire BiGG universal reaction pool. The former has a relatively small size (200-800 reactions depending on models) per GEM, while the latter has almost 17,000 reactions. A schematic plot of this validation is provided in the updated Fig. 2a (attached as Fig. R8a in this letter). The results of this validation can be found in the updated Fig. 2f-m (attached as Fig. R8f-m). CHESHIRE achieves the best performance over the other three methods when adding the top 25, top 50, top 100, and top N reaction with the highest confidence scores for both types of reaction pools (here N is the number of artificially removed reactions). We have updated the corresponding text in the revised manuscript:

"In the second type of internal validation, every step remains the same except that the testing set was not mixed with its derived negative reactions but with real reactions from a universal database." (see Page 8, Lines 141-143)

"To perform the second type of internal validation, we tested CHESHIRE on the same BiGG GEMs (with 90% training and 10% testing) with genus-specific and the universal BiGG reaction pools (Supplementary Information Section 4.2 and 4.3). The former has a relatively small size (200-800 reactions) per GEM, while the latter has almost 17,000 reactions. Since the number of reactions with similar biochemistry mechanisms and thus nearly identical confidence scores scale up with the size of candidate reaction pool, a loose threshold of 0.5 may still predict hundreds or thousands of candidates as missing reactions. Instead of using a fixed cutoff threshold, we added the top 25, 50, 100, and N reactions with the highest confidence scores (N is the number of artificially removed reactions). We found that CHESHIRE achieves the highest recovery rate at the four cutoffs for both types of reaction pools (Fig. 2f-m). Notably, by adding the top 25 reactions from the genus-specific reaction pools, CHESHIRE identifies more than 40% (on average) artificially removed reactions, significantly outperforming the other three methods (Fig. 2f; $P < 10^{-16}$, two-sided paired-sample t-test). While the performance of CHESHIRE declines when more reactions were added, it is still significantly better than the second-best method C3MM at the top N cutoff (Fig. 2i; $P < 0.05$). Furthermore, as expected, using the entire BiGG universal reaction pool would undermine the performances of recovery rate for all the methods. CHESHIRE nevertheless accomplishes the best performance compared to NHP and NVM (Fig. 2j-m; $P < 10^{-16}$)." (see Page 9-10, Lines 162-178)

Point 3.9. 6) For external validation, the best results were achieved when 500 reactions were added with CHESHIRE, but 500 gap-filled reactions are quite a lot given that models usually have 1500-2000 reactions. Manual gap filling processes are usually adding much fewer reactions (10-100).

Response: We thank Reviewer #3 for this critical comment. The reviewer is correct that CHESHIRE added many more reactions than manual gap-filling processes. However, manual gap-filling typically requires genetic or phenotypic data to guide the process. By contrast, CHESHIRE does not require experimental data as a priori knowledge. It is unfair to directly compare the number of reactions added by CHESHIRE and manual gap-filling methods. We agree with the reviewer that adding 500 reactions may be too much and risk of incorporating false positives. This comment is essentially the same as Point 1.11 raised by Reviewer 1 and Point 2.4 raised by Reviewer 2. Please see our responses to both comments for details. Briefly, we have successfully reduced the number of added reactions from 500 to 200 by using a similarity score (indicating the averaged similarity between a candidate reaction and those in the draft models) to automatically trim the candidate reactions. We have updated our approach for ranking reactions in the revised manuscript (see Page 11, Lines 189-197):

"Briefly, CHESHIRE is trained on the entire reaction set of a draft GEM, and candidate reactions (taken from a reaction pool, e.g., the BiGG database [30]) are ranked based on both confidence and similarity scores. The confidence score, returned by CHESHIRE, quantifies the probability of a candidate reaction being present in the GEM. The similarity score measures the maximum correlation between a candidate reaction and all existing reactions in the GEM. Given our rationale that dissimilar reactions are more likely to be functionally complementary to the existing ones, all candidate reactions whose confidence scores ≥ 0.9995 are ranked by their similarity scores (least similar to most similar). The top 200 reactions are added to the draft GEM to produce a gap-filled GEM (Supplementary Information Section 6.1)."

Point 3.10. How did the author assess false positives added in CHESHIRE-500? E.g., how many fermentation products were added, which were not observed experimentally?

Response: We thank Reviewer #3 for this critical comment. To assess the false positives added in the top 200 CHESHIRE-predicted reactions, we have counted the number of fermentation products and amino acids that are added by CHESHIRE but not observed experimentally. For the test of fermentation products, CHESHIRE adds a phenotype (draft models predicted a negative phenotype) in 27 simulations (a combination of model and metabolite), where 15 are true positives and 12 are false positives. For the test of amino acids, all 67 simulations that predicted a gain of phenotype are true positives (0 false positives). These results suggest that the risk of introducing false-positive phenotypes may depend on the model and phenotype. We have updated the corresponding text in the revised manuscript (see Page 14-15, Lines 266-272):

“To assess the false positives added by CHESHIRE-200, we counted the number of fermentation products and amino acids that were added by CHESHIRE but not observed experimentally. For the fermentation test, CHESHIRE adds a phenotype in 27 simulations (i.e., combinations of genome and metabolite), where 15 are true positives and 12 are false positives. For the amino acid test, all 67 simulations that predicted a gain of phenotype are true positives. These results suggest that the risk of introducing false positive phenotypes may depend on the model and phenotype.”

Point 3.11. Is CHESHIRE biased towards scoring specific types of reactions higher than others (e.g. transport reactions, reactions involving cofactors, etc?) It would be interesting to analyze what types of reactions are scored highest, and whether there is statistical overrepresentation of certain types (e.g. with procedures similar to pathway enrichment analysis).

Response: We thank Reviewer #3 for this excellent suggestion. We evaluated the types of reactions that are scored higher than the others. For each annotated enzymatic functional class (e.g., hydratase), we plotted the distributions of the ranks of all reactions whose enzymes belong to this class across all gap-filled GEMs. Then we ranked these functional classes by the distribution median (see Fig. R6, corresponding to Fig. S8 in the revised SI). For the majority of functional classes, there is a huge variability of the ranks among all its reactions, suggesting that the reactions with higher scores are context (GEM)-dependent and cannot be predicted from enzymatic functional class as a priori knowledge. Still, catalase, hydratase, dinucleosidetriphosphatase, cyclase, and laminaribiase ranked the top for the models in the fermentation product test, while the top five functional classes for the models in the amino acid test are dinucleosidetriphosphatase, hydratase, urease, cyclase, and phosphotransacetylase. Combining the two datasets reveals that reactions catalyzed by dinucleosidetriphosphatase, hydratase, and cyclase are typically scored higher on average. Despite addition of a single transport reaction has been shown to fill the gap of fermentation products and amino acids (see Fig. R9, corresponding to Fig. 3 in the revised manuscript), the median ranking of transport reactions is >700 and there is no overall bias towards transport reactions. We have updated the corresponding text in the revised manuscript (see Page 15, Lines 272-277):

“We further assessed whether these false positives may be linked to the bias of CHESHIRE to score specific types of reactions higher than the others. For nearly all enzymatic functional classes (Supplementary Information Section 6.5), we found a huge variability in the rankings of reactions catalyzed by enzymes that belong to each individual class (Fig. S8). Relatively, reactions catalyzed by dinucleosidetriphosphatase, hydratase, and cyclase are scored higher on average.”

In the revised manuscript, we explicitly acknowledged the limitation that CHESHIRE risks of adding false positives in the 200 reactions and discussed future solutions for trimming reactions that are wrongly added. Please see our responses to Point 2.4 and Point 2.7 for details.

Point 3.12. Minor comments:

7) On page 5, it is written that the main limitation of C3MM is that it cannot predict unseen reactions, since it includes all candidate reactions obtained from a universal reaction pool. However, it seems like CHESHIRE has exactly the same limitation, as it requires candidate reactions as input and provides their scoring. This limitation should be explicitly stated and discussed.

Response: We thank Reviewer #3 for this critical comment. We admit that the term “unseen reactions” is

misleading and have removed it from the revised manuscript. The main limitation of C3MM is that the model has to be re-trained when dealing with a new reaction pool (since C3MM requires all candidate reactions to be present during training). On the other hand, CHESHIRE (and NHP, NVM) do not have this limitation. It only requires a metabolic network for training. After learning the model parameter, CHESHIRE can be applied to any reaction pool and return confidence scores for candidate reactions without re-training the model. We have updated the corresponding text in the revised manuscript (see Page 5, Lines 69-72):

“C3MM has an integrated training-prediction process, which includes all candidate reactions (obtained from a reaction pool) during training. Hence, it has limited scalability (i.e., it cannot handle large reaction pools), and the model has to be re-trained for each new reaction pool.”

Point 3.13. 8) Page 15, it is written that “lactate consumption is preferred over production as it increases maximum growth rate”. How is growth rate included into the CHESHIRE pipeline, so that the authors can make this suggestion?

Response: We thank Reviewer #3 for this valuable comment. CHESHIRE *per se* does not compute growth rate. Given a draft or gap-filled GEM, optimization of its growth rate (or biomass production rate) is an essential intermediate step for the prediction of fermentation products (see Supplementary Information Section 6.3 for details). By optimizing growth, the influx of nutrients from the environment is determined and the quantitative nutrient utilization patterns impact the prediction results. *Anaerobutyricum hallii* has lactate dehydrogenase and can theoretically produce or use lactate for growth (the culture medium contains lactate). For the draft model, growth optimization favors lactate production because growth rate does not increase by utilizing lactate. For the draft model with 200 reactions predicted by CHESHIRE, growth optimization, however, favors lactate consumption which increases the growth rate. We have explained how growth optimization determines lactate consumption/production in the revised manuscript (see Page 14, Lines 255-261):

“For example, the draft GEM of Anaerobutyricum hallii has lactate dehydrogenase and can theoretically produce or use lactate for growth when lactate is present in the culture medium. The draft model predicts lactate production, since lactate utilization is dispensable for maximal biomass production. However, this prediction contradicts experimental data. CHESHIRE-200 fills this gap by adding two NAD(P)H-mediated redox reactions (Fig. 3k) that enable maximization of growth rate by consuming lactate.”

Point 3.14. Are the reports that the bacteria under study (*Anaerobutyricum hallii*) can consume lactate?

Response: We thank Reviewer #3 for this comment. Yes, our prediction of lactate consumption is supported by previous reports [10, 11]. We have cited experimental evidences of lactate consumption by *Anaerobutyricum hallii* in the revised manuscript (see Page 14, Lines 261-263):

“Supported by previous reports of lactate consumption [42, 43], this example shows that CHESHIRE can identify missing reactions that have consequences on distant fermentation pathways via a global and systematic effect.”

Point 3.15. 9) Figure 1, CHESHIRE schematic should include details about input candidate reaction pools and whether positive/negative labels are provided as well.

Response: We thank Reviewer #3 for pointing this out. We have updated Fig. 1 to indicate the candidate reaction pool (during prediction) and true labels (during training) (attached as Fig. R7).

Point 3.16. 10) Page 28, legend to Figure 1 g: “We identified gaps of intact draft GEMs by comparing model predictions with phenotypic data” – this sentence should be rephrased for clarity. I guess what is meant that the CHESHIRE predictions of filled gaps were assessed by comparing the gap-filled model performance to the original model in terms of fermentation reactions (phenotypic data), or something like that.

Response: We thank Reviewer #3 for this comment. Yes, we meant gap-filled GEMs by CHESHIRE have

better performance than draft GEMs in terms of phenotypic prediction. We have rephrased that sentence in the revised manuscript (see Page 10-11, Lines 183-186):

“Compared to internal validation that tests the predictions by using artificially removed reactions as the ground truth, external validation tests whether gap-filled GEMs by CHESHIRE has improved performance compared to draft GEMs in terms of their predictions of phenotypic data (Fig. 3a, Supplementary Information Section 6).”

Point 3.17. 11) Figure 3 b-i, why are performance metric separated for NHP and CHESHIRE? It seems like all the methods could be put together on the four plots (AUROC, Recall, Precision, F1 Score).

Response: We thank Reviewer #3 for this valuable comment. We have combined the results of NHP and CHESHIRE in one plot for a performance metric. Please see the updated Fig. 3b-i (attached as Fig. R9b-i).

Point 3.18. 12) Page 14, it is written that mean performance significantly increase. If the word “significant” is used, the p-value and the name of the statistical test should be provided as well.

Response: We thank Reviewer 3 for pointing this out. We have added the p-value and the name of the statistical test. We have updated the corresponding text in the revised manuscript:

“To the contrary, CHESHIRE-200 increases the mean performances significantly (Fig. 3b-e, Fig. S5) and, in particular, the F1 score for 11 of the 24 draft GEMs ($P < 0.01$, two-sided paired-sample t-test).” (see Page 12, Lines 214-217)

“Finally, we demonstrated that the improved performance is not simply due to more reactions by showing a significantly better performance of CHESHIRE-200 than that of Random-200 ($P < 0.05$, two-sided paired-sample t-test).” (see Page 12, Lines 220-222)

“Similar to the fermentation test described above, CHESHIRE-200 outperforms Random-200 (F1 score: $P < 10^{-5}$, two-sided paired-sample t-test), and NHP-200 shows no improvement at all (Fig. 3f-i, Fig. S5).” (see Page 13, Lines 235-237)

Point 3.13. 13) Typos:

- a. Page 1 Abstract: “spctral“
- b. Page 4 “involving → involved“
- c. Page 5 “scalibility“

Response: We thank Reviewer #3 for pointing out those typos, which have been corrected in the revised manuscript.

Finally, we thank Reviewer #3 again for reviewing our manuscript and her/his very insightful and constructive comments, which have helped us significantly improve the quality of our manuscript. We hope our responses above have addressed all her/his concerns in a satisfactory manner.

References

- [1] Stefanía Magnúsdóttir, Almut Heinken, Laura Kutt, Dmitry A Ravcheev, Eugen Bauer, Alberto Noronha, Kacy Greenhalgh, Christian Jäger, Joanna Baginska, Paul Wilmes, et al. Generation of genome-scale metabolic reconstructions for 773 members of the human gut microbiota. *Nature biotechnology*, 35(1):81–89, 2017.
- [2] Zachary A King, Justin Lu, Andreas Dräger, Philip Miller, Stephen Federowicz, Joshua A Lerman, Ali Ebrahim, Bernhard O Palsson, and Nathan E Lewis. Bigg models: A platform for integrating, standardizing and sharing genome-scale models. *Nucleic Acids Research*, 44(D1):D515–D522, 2016.
- [3] Nhung Pham, Ruben GA van Heck, Jesse CJ van Dam, Peter J Schaap, Edoardo Saccenti, and Maria Suarez-Diez. Consistency, inconsistency, and ambiguity of metabolite names in biochemical databases used for genome-scale metabolic modelling. *Metabolites*, 9(2):28, 2019.
- [4] Samuel MD Seaver, Filipe Liu, Qizhi Zhang, James Jeffryes, José P Faria, Janaka N Edirisinghe, Michael Mundy, Nicholas Chia, Elad Noor, Moritz E Beber, et al. The modelseed biochemistry database for the integration of metabolic annotations and the reconstruction, comparison and analysis of metabolic models for plants, fungi and microbes. *Nucleic acids research*, 49(D1):D575–D588, 2021.
- [5] Johannes Zimmermann, Christoph Kaleta, and Silvio Waschina. gapseq: Informed prediction of bacterial metabolic pathways and reconstruction of accurate metabolic models. *Genome biology*, 22(1):1–35, 2021.
- [6] Samir Giri, Leonardo Oña, Silvio Waschina, Shraddha Shitut, Ghada Yousif, Christoph Kaleta, and Christian Kost. Metabolic dissimilarity determines the establishment of cross-feeding interactions in bacteria. *Current Biology*, 31(24):5547–5557, 2021.
- [7] Yashu Liu, Shuang Qiu, Ping Zhang, Pinghua Gong, Fei Wang, Guoliang Xue, and Jieping Ye. Computational drug discovery with dyadic positive-unlabeled learning. In *Proceedings of the 2017 SIAM International Conference on Data Mining*, pages 45–53. SIAM, 2017.
- [8] Naganand Yadati, Vikram Nitin, Madhav Nimishakavi, Prateek Yadav, Anand Louis, and Partha Talukdar. Nhp: Neural hypergraph link prediction. In *Proceedings of the 29th ACM International Conference on Information & Knowledge Management*, pages 1705–1714, 2020.
- [9] Govind Sharma, Prasanna Patil, and M Narasimha Murty. C3mm: clique-closure based hyperlink prediction. In *Proceedings of the 29th International Conference on International Joint Conferences on Artificial Intelligence*, pages 3364–3370, 2020.
- [10] Sudarshan A Shetty, Simone Zuffa, Thi Phuong Nam Bui, Steven Aalvink, Hauke Smidt, and Willem M De Vos. Reclassification of *Eubacterium hallii* as *Anaerobutyricum hallii* gen. nov., comb. nov., and description of *Anaerobutyricum soehngeni* sp. nov., a butyrate and propionate-producing bacterium from infant faeces. *International Journal of Systematic and Evolutionary Microbiology*, 68(12):3741–3746, 2018.
- [11] Sylvia H Duncan, Petra Louis, and Harry J Flint. Lactate-utilizing bacteria, isolated from human feces, that produce butyrate as a major fermentation product. *Applied and environmental microbiology*, 70(10):5810–5817, 2004.

Tables

BiGG Model	Recon3D	iCHOv1	iLB1027_lipid	iCHOv1_DG44	RECON1
# Metabolites	5,835	4,456	2,172	2,751	2,766
# Reactions	10,600	6,663	4,456	3,942	3,741

Table R1: Statistics of the five largest BiGG models.

BiGG Model	Recon3D	iCHOv1	iLB1027_lipid	iCHOv1_DG44	RECON1
C3MM	2,871.36	1,456.44	636.37	441.51	444.37
NHP	321.38	175.88	91.14	86.18	86.28
CHESHIRE	211.15	109.55	63.30	45.04	42.83

Table R2: Comparison of running time (in second) of CHEHISRE, NHP, and C3MM on the five largest models from the BiGG database. The running time obtained is for the first type of internal validation in a Mactonish machine with Apple M1 Pro chip and 32 GB memory. This table is added as Table S4 of the revised Supplementary Information.

Hyperparameter	Universal	Grid Search Range
Embedding Dimension	256	{64, 128, 256}
Embedding Dimension	128	{64, 128, 256}
Chebyshev Filter Size	3	{2,3,4}
Dropout Probability	0.1	{0.1, 0.2}
Learning rate	0.01	{0.01, 0.001}

Table R3: Universal hyperparameter set and grid search range in the comparison test for CHESHIRE.

BiGG Model	iAF1260b	iAB_RBC_283	iYS854	iYO844	iSB619
Universal	0.8335	0.7972	0.8583	0.8935	0.8459
Grid Search	0.8439	0.7991	0.8639	0.8902	0.8518

Table R4: Average AUROC on the five randomly selected BiGG models under two settings using CHESHIRE.

Figures

Fig. R1: Internal validation using artificially introduced gaps on AGORA GEMs of gut bacteria. **a-d.** Boxplots of the performance metrics (AUROC, Recall, Precision, and F1 score) calculated on 818 AGORA GEMs (each dot represents a GEM) for CHESHIRE vs. NHP, C3MM, and NVM. Two-sided paired-sample t-test: $***P < 10^{-18}$. This figure is the same as Supplementary Information Fig. S4.

Fig. R2: External validation evaluated using overall accuracy and Matthew's correlation coefficients. **(a, b)** The fermentation metabolite test (24 bacterial GEMs). **(c, d)** The amino acid test (25 bacterial GEMs). Each dot represents a GEM. CarveMe: CarveMe-reconstructed GEMs; NHP-200: draft models plus 200 NHP-predicted missing reactions; CHESHIRE-200: draft models plus 200 CHESHIRE-predicted missing reactions; Random-200: draft models plus 200 randomly selected reactions (performance averaged over 3 Monte Carlo runs). Two-sided paired-sample t-test: n.s., not significant; * $P < 0.05$; ** $P < 0.01$; *** $P < 10^{-5}$. This figure is the same as Supplementary Information Fig. S5.

Fig. R3: The fermentation metabolite test (external validation) using ModelSEED-reconstructed draft GEMs. **a-d.** Boxplots of the performance metrics (AUPRC, Recall, Precision, and F1 score) calculated on 24 bacterial GEMs for CHESHIRE-200 (draft models plus 200 CHESHIRE-predicted missing reactions) vs. ModelSEED (ModelSEED-reconstructed GEMs), NHP-200 (draft models plus 200 NHP-predicted missing reactions), and Random-200 (draft models plus 200 randomly selected reactions; performance averaged over 3 Monte Carlo runs). Two-sided paired-sample t-test: n.s., not significant; $*P < 0.05$; $**P < 0.01$; $***P < 10^{-5}$. This figure is the same as Supplementary Information Fig. S6.

Fig. R4: Testing CHESHIRE's performance with genus-specific BiGG reaction pools on the fermentation metabolite production. CarveMe: draft models reconstructed from the CarveMe pipeline; CHESHIRE-200: draft CarveMe models plus 200 reactions predicted by CHESHIRE. Each dot in a boxplot represents a GEM. ns: non-significant as determined by the two-sided paired-sample t-test. See Supplementary Information Section 4.3 for details of constructing the genus-specific reaction pools.

Fig. R5: Performance evaluation of CHESHIRE on filling the gaps in (a) growth phenotype and (b) gene essentiality. Each dot in a boxplot represents a GEM. CarveMe: draft models reconstructed from the CarveMe pipeline; CHESHIRE-200: draft models plus 200 missing reactions predicted by CHESHIRE; NHP-200: draft models plus 200 missing reactions predicted by NHP; Random-200: draft models plus 200 reactions after randomization. n.s.: non-significant as determined by the two-sided paired-sample t-test. This figure is the same as Supplementary Information Fig. S7.

Fig. R6: Reaction rankings categorized by enzymatic functional classes. Each dot represents a specific reaction and all dots for each boxplot represent all reactions catalyzed by enzymes of a specific functional class. Panel **a** was drawn using reaction rankings from GEMs in the fermentation product test and panel **b** was drawn using reaction rankings from GEMs in the amino acid test. This figure is the same as Supplementary Information Fig. S8.

Fig. R7: CHESHIRE workflow. **a.** Schematic representation of a metabolic network. **b.** Hypergraph representation of the metabolic network. The hypergraph is undirected where each hyperlink connects metabolites that participate the same reaction. **c.** Negative sampling of the metabolic network. Solid and dashed boxes represent positive and negative reactions (e.g., N1, N2), respectively. **d.** Decomposed graph of the metabolic network, where each reaction (either positive or negative) is treated as a fully connected subgraph (solid and dashed lines represent positive and negative reactions, respectively). **e.** The architecture of CHESHIRE during training. The deep neural network takes the incidence matrix and the decomposed graphs as the input, and consists of an encoder layer, a Chebyshev spectral graph convolutional layer with K filters (resulting in K channels), a pooling layer with two pooling functions, and a final scoring layer. The output confidence scores are compared to the target scores for updating model parameters. The gray dots represent the hidden neurons. **f.** The architecture of CHESHIRE during prediction. The neural network takes the incidence matrix and a decomposed graph built from candidate reactions as input and outputs confidence scores for candidate reactions based on the trained model parameters. This figure is the same as Fig. 1 in the revised manuscript.

Fig. R8: Internal validation using artificially introduced gaps. **a.** Flowchart of internal validation. Two types of internal validation were performed. The former mixes artificially removed positive reactions and their derived negative reactions as candidate reactions, while the latter uses artificially removed positive reactions and real reactions from a universal reaction database as candidate reactions. **b-e.** Boxplots of the performance metrics (AUROC, Recall, Precision, and F1 score) calculated on 108 BiGG GEMs (each dot represents a GEM) for CHESHIRE vs. NHP, C3MM, and NVM. **f-i.** Reaction recovery rate of CHESHIRE vs. NHP, C3MM, and NVM with genus-specific reaction pools for gap-filling the BiGG GEMs by adding the top 25, 50, 100, and N reactions with the highest confidence scores (N is the number of artificially removed reactions). **j-m.** Reaction recovery rate of CHESHIRE vs. NHP and NVM with the entire BiGG universal reaction pool for gap-filling the BiGG GEMs by adding the top 25, 50, 100, and N reactions with the highest confidence scores. For **f-m**, only the BiGG models with over 100 reactions were tested, and C3MM was only validated on the genus-specific reaction pools due to the issue of scalability. Each data point represents the mean statistic over 10 Monte Carlo runs. Boxplot: central line represents the median, box limits represent the first and third quartiles, and whiskers extend to the smallest and largest values or at most to $1.5\times$ the interquartile range, whichever is smaller. Two-sided paired-sample t-test: $*P < 0.05$; $**P < 10^{-4}$; $***P < 10^{-16}$. This figure is the same as Fig. 2 in the revised manuscript.

Fig. R9: External validation by predicting metabolic phenotypes. **a.** Flowchart of external validation. The predicted phenotypes from CHESHIRE-gapfilled GEMs are validated by comparison to experimental observation. For phenotypes correctly predicted by gap-filled GEMs but missed by draft GEMs, we also identify the causal reactions from CHESHIRE-predicted set that improve the phenotypic prediction using Mixed Integer Linear Programming (MILP). **b-i.** Performance (AUPRC, Recall, Precision, and F1 score) of CHESHIRE and NHP in filling gaps in **(b-e)** 24 bacterial GEMs for fermentation metabolite production and **(f-i)** 25 bacterial GEMs for amino acid secretions. NVM was not included here due to its poor performance in internal validation. C3MM was not considered either, because it requires infeasible computational time for such a large candidate reaction pool (with almost 17,000 reactions). “CarveMe” represents the draft models reconstructed from CarveMe. “NHP-200” and “CHESHIRE-200” represent draft models plus 200 reactions predicted by NHP and CHESHIRE, respectively. For “Random-200”, 200 randomly selected reactions from the universal BiGG database were added to the draft models. Boxplot: central line represents the median, box limits represent the first and third quartiles, and whiskers extend to the smallest and largest values or at most to $1.5\times$ the interquartile range, whichever is smaller. Two-sided paired-sample t-test: n.s., not significant; $*P < 0.05$; $**P < 0.01$; $***P < 10^{-5}$. **j-m.** Examples of CHESHIRE-predicted reactions (red arrows) that causally gap-fill the observed phenotypes of acetate production (j), lactate production (k), and amino acid secretions (l and m). Abbreviations of cofactors: adenosine triphosphate (ATP); adenosine diphosphate (ADP); adenosine phosphate (AMP); phosphate (Pi); inorganic pyrophosphate (PPi); Coenzyme A (CoA); oxidized/reduced nicotinamide adenine dinucleotide (NAD^+/NADH); oxidized/reduced nicotinamide adenine dinucleotide phosphate ($\text{NADP}^+/\text{NADPH}$). This figure is the same as Fig. 3 in the revised manuscript.

Reviewers' Comments:

Reviewer #1:

Remarks to the Author:

Remarks to the Author:

The authors have sufficiently improved their paper, in response to the comments made. I enjoyed reading it.

although there are some major concerns remained with the methodology itself. the main concern would be the number of reactions that must be added to a network to fill its gaps (previously 500 reactions - reduced to 200 reactions after the first revision). in this regard, there are several available gap-filling approaches that are capable of filling networks gaps by adding less than 50 reactions regardless of network size and the species, But all are template based and computationally expensive. current approach would be highly beneficial when it comes to reconstructing GEMs for non-model organisms lacking a previous template for gap-filling. The following comments may help to improve the methodology to some extent.

1- a downstream workflow to assess the necessity of reactions that are added to networks to fill its gaps. this could be made by running a simple reaction essentiality analysis (both for Biomass objective function and targeted fermentation metabolites) on a set of 200 reactions that are added to the network. this may infer some of the mentioned reactions are not essential either for growth or for enhancing prediction on fermentation profile and thus may result in a reduced number of gap reactions.

2- given that NADH and NADPH are the main redox cofactors of catabolic and anabolic pathways respectively, it might be helpful to consider a weighting score for reactions with different cofactors as further selection criteria, this may potentially help to reduce the number of added reactions by reducing the number of redundant reactions.

3- checking the directionality of CHESHIRE-200 is a simple and doable analysis and may help to make the number of gaps fall into a reasonable range (less than 100). this could be done by checking whether any two reactions within CHESHIRE-200 have the same stoichiometry but different directionality. any two reactions that meet these conditions could be merged into a bi-directional reaction.

Reviewer #2:

Remarks to the Author:

I appreciate the comprehensive response of the authors. They have addressed my concerns.

Reviewer #3:

Remarks to the Author:

Chen, Liao and Liu present a revised manuscript describing a method to predict missing reactions in genome-scale metabolic networks (GEMs) through deep learning based on CHEbyshev Spectral Hyperlink pREdictor (CHESHIRE). In this revised version, the authors provide a more thorough explanation of the method and its inputs, outputs and limitations; expand the validation datasets, and investigate the types of reactions and models for which the method works better or worse. This revised manuscript is a substantial improvement over the previous version, and my comments and concerns raised before have been addressed in a satisfactory manner. It is important to note that the authors now better clarified the difference between their approach and supervised optimisation-based gap filling approaches. Although unsupervised addition of hundreds of reactions is rather questionable for biological applications, I find that the application of graph neural networks in genome-scale metabolic modelling will be of high interest to a broader audience working on machine learning

applications in biology, mathematical modelers, and system biologists, and could serve as a stepping stone for further development of methods and applications in this field.

Response to Reviewer #1

Point 1.0. The authors have sufficiently improved their paper, in response to the comments made. I enjoyed reading it. although there are some major concerns remained with the methodology itself. the main concern would be the number of reactions that must be added to a network to fill its gaps (previously 500 reactions - reduced to 200 reactions after the first revision). in this regard, there are several available gap-filling approaches that are capable of filling networks gaps by adding less than 50 reactions regardless of network size and the species, But all are template based and computationally expensive. current approach would be highly beneficial when it comes to reconstructing GEMs for non-model organisms lacking a previous template for gap-filling. The following comments may help to improve the methodology to some extent.

Response: We thank Reviewer #1 for reviewing our manuscript again. Next, we address each of her/his remaining comments in order.

Point 1.1. 1- a downstream workflow to assess the necessity of reactions that are added to networks to fill its gaps. this could be made by running a simple reaction essentiality analysis (both for Biomass objective function and targeted fermentation metabolites) on a set of 200 reactions that are added to the network. this may infer some of the mentioned reactions are not essential either for growth or for enhancing prediction on fermentation profile and thus may result in a reduced number of gap reactions.

Response: We are grateful to the reviewer for suggesting this improvement. Our draft CarveMe or ModelSeed models were generated using standard pipelines and we employed growth phenotype for gap-filling within the CarveMe or ModelSeed workflow. This decision was made because CHESHIRE was not designed for gap-filling growth phenotype and, as such, cannot guarantee the growth of gap-filled GEMs, which is necessary for all metabolic phenotype simulations. Given that the draft GEMs can grow before CHESHIRE-guided gap-filling procedure starts, none of the candidate reactions will be essential for the growth phenotype. Moreover, the identification of essential reactions for gap-filled metabolite production is a built-in function of CHESHIRE. Essentially, CHESHIRE-200 screened for 236 secretable metabolites and assessed whether the addition of 200 reactions enabled the production of any metabolites that could not be produced by the draft GEMs before gap-filling. The essential reactions for any potential metabolic gaps were also reported by CHESHIRE (see README page of our Github repository: <https://github.com/canc1993/cheshire-gapfilling>). We have updated the corresponding text in the revised manuscript (see Page 17-18, Line 323-328):

"In this study, we have taken initial steps towards further reduction of the number of CHESHIRE-predicted reactions. First, CHESHIRE-200 conducted a comprehensive search of 236 metabolites to identify metabolic phenotypes that can be potentially gap-filled by adding 200 reactions (Supplementary Information Section 6.3). The tool also provides information on the essential reactions for each potential gap, allowing users to focus on the most promising reactions without being overwhelmed by all the 200 reactions."

Point 1.2. 2- given that NADH and NADPH are the main redox cofactors of catabolic and anabolic pathways respectively, it might be helpful to consider a weighting score for reactions with different cofactors as further selection criteria, this may potentially help to reduce the number of added reactions by reducing the number of redundant reactions.

Response: We thank Reviewer #1 for this excellent suggestion. We have taken the initial steps towards exploring the possibility of prioritizing reactions based on the cofactors they involve. For each draft GEM, we focused on 15 different cofactors and ranked them by the number of reactions they are involved in the GEM. Then we tested whether excluding reactions containing any cofactor or those with the least prevalent cofactors can improve the gap-filling performance in the two datasets we introduced for external validation. We did not observe a consistent trend regarding the performance change when reactions involving rare cofactors are excluded: our approach improved performance in one dataset and decreased it in the other (compare CHESHIRE-100, CHESHIRE-100-Top3, CHESHIRE-100-Top2, CHESHIRE-100-Top1, and CHESHIRE-100-NoCF in Fig. R1b,d). However, when candidate reactions are not allowed to contain any cofactor, we were able to reduce the number of reactions to 100 while maintaining significant performance improvement between gap-filled and draft GEMs (compare CHESHIRE-100-NoCF and CarveMe in Fig. R1b,d). Further reduction of the number of reactions to 50 does not yield any significant improvement for both datasets (compare CHESHIRE-100-NoCF and CarveMe in Fig. R1a,c). Our preliminary analysis suggests that further studies on the distribution of potential gaps among different co-factor-based reactions may open up new avenues for metabolic network gap-filling. We have added Fig R1 as Fig. S9 to the revised SI, and added a paragraph to describe our approaches and findings in the revised main text (see Page 18, Line 328-337)

“Moreover, we explored the feasibility of prioritizing reactions with different cofactors (e.g., NADH) based on their prevalence in draft GEMs (Supplementary Information Section 6.1). We found that excluding candidate reactions involving less prevalent cofactors led to improved performance for the secretion of fermentation products, but decreased performance for the dataset of amino acid secretions (Fig. S9). These contrasting trends suggest that the relationship between missing reactions and cofactors may depend on the secreted products and their biosynthetic pathways. Importantly, the co-factor-based strategy enabled CHESHIRE to reduce the number of added reactions to 100, while still significantly improving gap-filled GEMs over draft GEMs. Thus, our preliminary analysis highlights a promising future direction for metabolic network gap-filling.”

Point 1.3. 3- checking the directionality of CHESHIRE-200 is a simple and doable analysis and may help to make the number of gaps fall into a reasonable range (less than 100). this could be done by checking whether any two reactions within CHESHIRE-200 have the same stoichiometry but different directionality. any two reactions that meet these conditions could be merged into a bi-directional reaction.

Response: We thank Reviewer #1 for this insightful comment. Unfortunately, all BiGG universal reactions (which can be downloaded from http://bigg.ucsd.edu/data_access) are unique and bidirectional. Therefore, reactions cannot be merged based on differences in their directionality. We have revised our manuscript to emphasize this point (see Page 19, Lines 350-354):

“Neither the BiGG universal reaction database nor the current version of CHESHIRE considers reaction directionality. Therefore, CHESHIRE cannot fill gaps caused by wrong directions or reduce the number of added reactions by merging reactions with the same stoichiometry but different directionality. Further studies are warranted to incorporate reaction directionality in the CHESHIRE framework. ”

Finally, we would like to point out that we changed our title to “Teasing out Missing Reaction in Genome-scale Metabolic Networks through Hypergraph Learning.” The reason for the change is that our method, CHESHIRE, is developed based on the hypergraph representation of metabolic networks. We believe that using the term “hypergraph learning” more accurately reflects the methodology. Furthermore, the previous title “Teasing out Missing Reactions in Genome-scale Metabolic Networks through Graph Convolutional Networks” had two “networks”, which could potentially cause confusion. We thank Reviewer #1 again for reviewing our manuscript and her/his very insightful and constructive comments, which have helped us significantly improve the quality of our manuscript. We hope our responses above have addressed all her/his concerns in a satisfactory manner.

Response to Reviewer #2

Point 2.0. I appreciate the comprehensive response of the authors. They have addressed my concerns.

Response: We thank Reviewer #2 for reviewing our manuscript again.

Response to Reviewer #3

Point 3.0. Chen, Liao and Liu present a revised manuscript describing a method to predict missing reactions in genome-scale metabolic networks (GEMs) through deep learning based on CHEbyshev Spectral Hyperlink pREdictor (CHESHIRE). In this revised version, the authors provide a more thorough explanation of the method and its inputs, outputs and limitations; expand the validation datasets, and investigate the types of reactions and models for which the method works better or worse. This revised manuscript is a substantial improvement over the previous version, and my comments and concerns raised before have been addressed in a satisfactory manner. It is important to note that the authors now better clarified the difference between their approach and supervised optimisation-based gap filling approaches. Although unsupervised addition of hundreds of reactions is rather questionable for biological applications, I find that the application of graph neural networks in genome-scale metabolic modelling will be of high interest to a broader audience working on machine learning applications in biology, mathematical modelers, and system biologists, and could serve as a stepping stone for further development of methods and applications in this field.

Response: We thank Reviewer #3 for reviewing our manuscript again and thoroughly summarizing our work.

Figures

Fig. R1: Comparison of CHESHIRE performance across various strategies for prioritizing cofactor-containing reactions using the fermentation metabolite test (24 bacterial GEMs) [28] in panels (a, b) and the amino acid test (25 bacterial GEMs) [29] in panels (c, d). The tested strategies include: CarveMe - GEMs reconstructed using CarveMe; CHESHIRE-50/100/200 - draft models plus 50/100/200 CHESHIRE-predicted reactions (all cofactors allowed); CHESHIRE-50/100/NoCF - draft models plus 50/100 CHESHIRE-predicted reactions that involve none of the 15 specified cofactors (see Section 6.1); CHESHIRE-50/100-Top1/Top2/Top3 - draft models plus 50/100 CHESHIRE-predicted reactions that do not involve any of the 15 specified cofactors or involve only the top 1/2/3 cofactors with the highest prevalence in the draft GEMs. Two-sided paired-sample t-test: n.s., not significant; $**P < 0.01$; $***P < 10^{-5}$.

Reviewers' Comments:

Reviewer #1:

Remarks to the Author:

Dear Authors,

I am grateful for your efforts to address my previous comments and for the significant improvements you have made to the manuscript. It is a pleasure to witness your commitment to enhancing the quality of your work.

Your diligent work is a testament to your dedication to scientific rigor, and I congratulate you on this excellent contribution to the field. I am grateful for the opportunity to review your manuscript, and I eagerly anticipate its publication in Nature Communications.

Response to Reviewer #1

Point 1.0. Dear Authors,

I am grateful for your efforts to address my previous comments and for the significant improvements you have made to the manuscript. It is a pleasure to witness your commitment to enhancing the quality of your work.

Your diligent work is a testament to your dedication to scientific rigor, and I congratulate you on this excellent contribution to the field. I am grateful for the opportunity to review your manuscript, and I eagerly anticipate its publication in Nature Communications.

Sincerely,

Response: We thank Reviewer #1 again for reviewing our manuscript and her/his very insightful comments, which have helped us a lot improve the quality of our work.